# HESSIAN-ENHANCED TOKEN ATTRIBUTION (HETA): INTERPRETING AUTOREGRESSIVE LLMS

**Vishal Pramanik**
Department of Computer & Information
Science & Engineering,
University of Florida
Gainesville, FL 32611, USA
vishalpramanik@ufl.edu

**Maisha Maliha**
School of Computer Science,
University of Oklahoma
Norman, OK 73019, USA
maisha.maliha-1@ou.edu

**Nathaniel D. Bastian**
Department of Electrical Engineering and Computer Science,
United States Military Academy
West Point, NY 10996, USA
nathaniel.bastian@westpoint.edu

**Sumit Kumar Jha**
Department of Computer & Information
Science & Engineering,
University of Florida
Gainesville, FL 32611, USA
sumit.jha@ufl.edu

## ABSTRACT

Attribution methods seek to explain language model predictions by quantifying the contribution of input tokens to generated outputs. However, most existing techniques are designed for encoder-based architectures and rely on linear approximations that fail to capture the causal and semantic complexities of autoregressive generation in decoder-only models. To address these limitations, we propose **Hessian-Enhanced Token Attribution (HETA)**, a novel attribution framework tailored for decoder-only language models. HETA combines three complementary components: a semantic transition vector that captures token-to-token influence across layers, Hessian-based sensitivity scores that model second-order effects, and KL divergence to measure information loss when tokens are masked. This unified design produces context-aware, causally faithful, and semantically grounded attributions. Additionally, we introduce a **curated benchmark dataset** for systematically evaluating attribution quality in generative settings. Empirical evaluations across multiple models and datasets demonstrate that HETA consistently outperforms existing methods in attribution faithfulness and alignment with human annotations, establishing a new standard for interpretability in autoregressive language models. Our code and dataset are available here.[1]

## 1 INTRODUCTION

As machine learning systems achieve increasingly high performance, they are being deployed in high-stakes domains such as healthcare, autonomous driving, and finance. However, despite their success, deep neural networks remain difficult to interpret due to their large parameter spaces, layered architectures, and nonlinear computations, earning them the reputation of "black box" models Benítez et al. (1997). This opacity can erode trust, impede debugging, and raise ethical or regulatory concerns. To address these challenges, the field of Explainable AI (XAI) emerged, with the goal of making model decisions more transparent, interpretable, and trustworthy.

A wide range of interpretability methods such as LIME Ribeiro et al. (2016), KernelSHAP Lundberg & Lee (2017), Integrated Gradients Sundararajan et al. (2017), Grad-CAM Selvaraju et al. (2017), and Layer-wise Relevance Propagation (LRP) Bach et al. (2015) have been developed under the classical feature attribution paradigm, which aims to quantify the contribution of input features to a model's output. Most of these methods are based on linear or first-order derivative approximations

---

[1]https://github.com/VishalPramanik/HETA.git

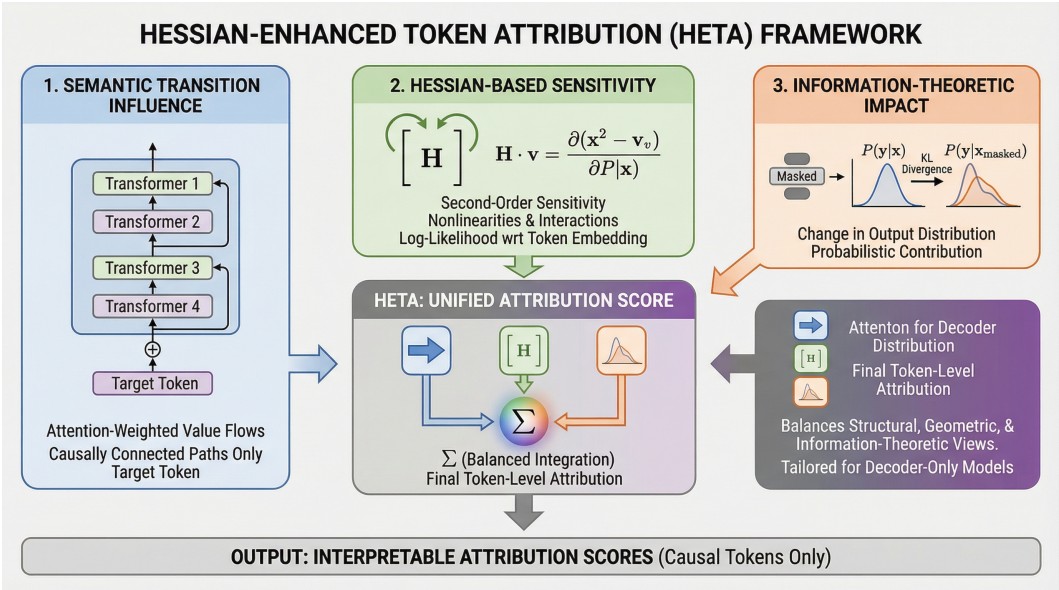

Figure 1: **Overview of HETA.** The pipeline (a) rolls out attention–value flows that end at the target token to form a causal gate over input tokens, (b) estimates token-level curvature via scalable Hessian–vector products to capture nonlinear interactions, and (c) measures KL-based information impact under token masking. The final attribution combines causal gating, curvature sensitivity, and information gain to produce target-conditioned, token-level explanations.

and assume local model linearity. However, this assumption often breaks down in the context of autoregressive language models, where token interactions are nonlinear and highly contextual. Despite their practical utility, these techniques frequently produce inconsistent attributions for the same input and model Hooker et al. (2019), casting doubt on their reliability. Although some efforts have introduced axiomatic foundations to formalize attribution Han et al. (2022); Bressan et al. (2024), a universally accepted definition of explanation quality remains elusive. Furthermore, these attribution methods have primarily been designed for encoder-based architectures, and recent work Zhao & Shan (2024) shows that directly applying them to decoder-only language models in generative tasks is non-trivial and often unfaithful. The discrepancy arises from architectural and functional differences, where encoder models leverage bidirectional attention and require a single attribution map, while decoder-only models generate outputs autoregressively and demand attribution at each token position. Figure 1 illustrates the complexity of the attribution task for a generative model, highlighting how input words contribute to the generation of a specific output word. Although model-agnostic approaches have been proposed for generative settings Zhao & Shan (2024), they typically ignore the dense semantic structure encoded in the internal layers of large language models Chen et al. (2024); Xu et al. (2020), thereby limiting their ability to capture deep token-level influence.

To address the shortcomings of gradient-based attribution methods in autoregressive models, we propose **Hessian-Enhanced Token Attribution (HETA)**, a framework tailored for decoder-only architectures. HETA combines semantic flow tracing, Hessian-based sensitivity, and KL-based information loss to yield faithful, token-level attributions. By modeling attention-weighted value flow and capturing second-order and informational effects, HETA offers a principled and robust alternative that respects the causal and contextual structure of generative language models. The key contributions of this work are:

- We propose **Hessian-Enhanced Token Attribution (HETA)**, a novel attribution method for decoder-only language models that integrates semantic flow for causal directionality, Hessian-based sensitivity for capturing second-order interactions, and KL-divergence for quantifying information-theoretic impact.

- We construct and release a new curated dataset specifically designed for evaluating token-level attributions in autoregressive generation tasks, enabling systematic assessment of attribution faithfulness, robustness, and human alignment.
- We conduct extensive experiments across four diverse decoder-only models and a broad suite of strong attribution baselines. Results show that HETA consistently outperforms existing methods in faithfulness, robustness, and semantic alignment, while exhibiting remarkable stability under both decoding hyperparameter changes and syntactic rephrasings.

## 2 MOTIVATION

Understanding which input tokens influence a generative language model's output is central to interpretability. However, existing attribution methods based on attention weights or first-order sensitivity are fundamentally limited in faithfully capturing token-level influence.

Attention-based methods Abnar & Zuidema (2020), while widely used, are not reliable indicators of causal influence. Attention weights reflect where the model attends, not what actually affects the output, and can often be perturbed without significantly altering predictions Jain & Wallace (2019). Moreover, attention mechanisms—especially when aggregated across layers or heads—often fail to account for indirect or multi-hop influence paths that propagate through residual connections and MLP layers Lu et al. (2021). In decoder-only models, attention is explicitly masked to prevent information flow from future tokens; yet, post-hoc attention aggregations that disregard this constraint may inadvertently assign importance to tokens that are not causally connected to the output. As a result, attention alone should not be treated as a faithful attribution signal.

First-order attribution methods, such as pointwise gradients, Input×Gradient Shrikumar et al. (2017), and DIG Sanyal & Ren (2021), measure the local linear sensitivity of the model's output with respect to its inputs. While computationally efficient, these methods can entirely miss meaningful influence in regions where the gradient vanishes but the function remains sensitive to finite perturbations. Integrated Gradients (IG) Sundararajan et al. (2017) partially addresses this by accumulating gradients along a path from a baseline to the input, but its attributions depend heavily on the choice of baseline and path, and can underrepresent influence near sharp transitions or in highly nonlinear regimes.

To illustrate this limitation, consider the model's prediction function $f(x) = \log P(x_T \mid x_{<T})$, where $x$ denotes the concatenated input embeddings (or intermediate hidden states) and $T$ is the target position. Even if $f$ is differentiable, it is possible for the partial derivative with respect to some input dimension to be zero while a finite perturbation in that direction still causes a nontrivial change in $f(x)$. More formally, for some $i \in \{1, \ldots, n\}$, we may have $\frac{\partial f(x)}{\partial x_i} = 0$ but $f(x + \epsilon e_i) \neq f(x)$ for some $\epsilon > 0$, indicating that the gradient fails to detect influence that manifests at higher orders.

This phenomenon is captured by the second-order Taylor expansion:

$$f(x) = f(x_0) + \nabla f(x_0)^\top (x - x_0) + \tfrac{1}{2}(x - x_0)^\top \nabla^2 f(\xi)(x - x_0),$$

for some $\xi$ on the segment between $x_0$ and $x$. When the gradient at $x_0$ vanishes, changes in the function are driven entirely by the curvature encoded in the Hessian, and the contribution scales quadratically with the perturbation norm. For example, in the smooth activation $f(x) = \log(1 + \exp(w^\top x + b))$, the gradient can be nearly zero in the saturated regime ($w^\top x + b \ll 0$), yet the function value still changes significantly for finite perturbations. In such cases, gradient-based attribution methods underestimate or entirely miss the relevant influence (see appendixA1 for details).

Recent studies like, ContextCite Cohen-Wang et al. (2024), TDD Feng et al. (2024), and *Peering into the Mind of LMs* Phukan et al. (2024) all aim to attribute outputs in generative language models, but each exhibits notable limitations. ContextCite relies on a sparse linear surrogate trained through extensive ablations, which makes it sensitive to redundancy and indirect dependencies, computationally expensive, and limited to sentence-level attribution without next-token conditioning. TDD projects hidden states through the model's output head (logit lens), conflating correlation with causation and producing saliency scores that are highly sensitive to the choice of target–alternative token pairs and vocabulary dynamics. *Peering* matches hidden states of generated answer tokens to context tokens using cosine similarity with layer-specific thresholds, a representation-matching approach that performs well primarily for verbatim spans but lacks robustness to paraphrasing, reordering, or indirect evidence chains.

These limitations, namely, the inability of attention to capture causal importance and the failure of first-order gradients to model non-linear sensitivity, underscore the need for more robust attribution frameworks. Existing methods remain unstable under decoding hyperparameter changes and syntactic rephrasings (see Section5). We propose **Hessian-Enhanced Token Attribution (HETA)**, which integrates causal semantic flow, second-order sensitivity, and output-aware information gain. Together, these components yield stable, faithful, and interpretable attributions for decoder-only LMs.

## 3 BACKGROUND

Understanding token-level influence in transformer models requires going beyond raw attention weights or local gradients. Two complementary strands of research have highlighted important limitations and proposed more robust alternatives. Kobayashi et al. (2020) demonstrated that attention weights alone are insufficient for faithful interpretation, as they neglect the scale of the value vectors being attended to. They proposed a norm-based approach that combines attention weights with the magnitude of the transformed value projections, offering a more accurate view of token influence within self-attention. This formulation captures not only alignment (via attention) but also semantic strength (via vector norms), leading to more faithful attributions.

Separately, Hessian-based sensitivity methods provide deeper insight into model behavior by accounting for second-order interactions between inputs and outputs. Specifically, the Hessian of the log-likelihood with respect to input embeddings, $H_T = \nabla_X^2 \log P(x_T \mid x_{<T})$, captures local curvature and reveals how token effects manifest in nonlinear regions of the model's decision surface. Unlike first-order methods which can fail in flat regions or under poor baseline selection, second-order methods remain informative even when gradients vanish. Prior studies (Dong et al. (2025), Alvarez-Melis & Jaakkola (2018), Dhamdhere et al. (2018)) support the use of Hessian-based approaches to uncover latent influences in deep architectures.

Together, these techniques underscore the importance of considering both semantic flow and higher-order sensitivity to capture faithful token attributions in transformer models.

## 4 OUR METHODOLOGY

We propose **Hessian-Enhanced Token Attribution (HETA)**, a principled framework that integrates these perspectives into a *unified influence decomposition*. Our central view is that token attribution in autoregressive models should estimate a token's *directional, target-conditioned causal contribution* to the log-likelihood of the current target token, incorporating both *semantic path dependencies* and *higher-order effects*. HETA achieves this through three complementary components: **(1) Semantic Transition Influence**, which captures how tokens propagate influence through compositional attention–value flows across layers, enforcing causal directionality toward the target. **(2) Hessian-Based Sensitivity**, which models second-order curvature of the log-likelihood surface with respect to token embeddings, capturing nonlinear and interaction effects. **(3) Information-Theoretic Impact**, which measures the change in predictive uncertainty when a token is masked, providing a probabilistic interpretation of its contribution. Together, these components form a mathematically grounded attribution score that balances structural, geometric, and information-theoretic perspectives on token influence.

Let $x_{1:T}$ be the input sequence with embeddings $\mathbf{X} = (\mathbf{e}_1, \ldots, \mathbf{e}_T) \in \mathbb{R}^{T \times d}$. A decoder-only model $f_\theta$ defines the conditional distribution over the *target* token:

$$P_\theta(x_T \mid x_{<T}) = \mathrm{Softmax}\big(f_\theta(\mathbf{X})\big). \tag{1}$$

Our goal is a nonnegative score $\mathrm{Attr}(x_i \to x_T)$ quantifying the contribution of token $x_i$ to predicting $x_T$. The score combines (i) semantic transition influence, (ii) Hessian-based sensitivity, and (iii) KL-based information loss.

**Semantic Flow for Causal Token Influence** To ensure *target-conditioned* causality, we trace attention-weighted value flow that *terminates at position $T$* under the decoder's causal mask. For each layer $l \in \{1, \ldots, L\}$ and head $h \in \{1, \ldots, H\}$, let $A^{(l,h)} \in \mathbb{R}^{T \times T}$ be the masked attention matrix, $V^{(l,h)} \in \mathbb{R}^{T \times d}$ the value vectors, and $W_O^{(l,h)} \in \mathbb{R}^{d \times d}$ the output projection. We compute a

target-conditioned attention rollout $\Phi^{(l,h)}(i \to T)$ (e.g., Abnar & Zuidema (2020)) that aggregates only paths ending at $T$. The semantic transition influence is

$$M_T[i] \;=\; \frac{1}{Z} \sum_{l=1}^{L} \sum_{h=1}^{H} \Phi^{(l,h)}(i \to T) \left\| V_i^{(l,h)} W_O^{(l,h)} \right\|_1, \qquad Z = \sum_{j=1}^{T} \sum_{l,h} \Phi^{(l,h)}(j \to T) \left\| V_j^{(l,h)} W_O^{(l,h)} \right\|_1.$$

Thus $M_T \in \mathbb{R}^T_{\geq 0}$ is simplex-normalized ($\sum_i M_T[i] = 1$) and assigns mass only to tokens with causal paths to $T$.

**Hessian-Based Sensitivity Analysis**   To capture second-order effects, we consider the Hessian of the target log-probability with respect to $\mathbf{X}$:

$$H_T \;=\; \nabla_{\mathbf{X}}^2 \log P_\theta(x_T \mid x_{<T}) \;\in\; \mathbb{R}^{(Td)\times(Td)}. \tag{2}$$

Explicitly forming $H_T$ is infeasible for realistic $T, d$. We therefore estimate block sensitivities via Hessian–vector products (HVPs) with Hutchinson estimators. Let $\Pi_i$ select token $i$'s $d$-dimensional block. With Rademacher vectors $r_k$ supported on that block, the sensitivity used in  equation 5 is

$$S_i^{(T)} \;\approx\; \frac{1}{m} \sum_{k=1}^{m} \left\| \Pi_i\, H_T\, (\Pi_i r_k) \right\|_1, \tag{3}$$

where each HVP is computed by Pearlmutter's trick; we optionally use a Gauss–Newton/Fisher surrogate for numerical stability. We report $m$, runtime, and memory in our experiments.

**KL Divergence for Information Contribution**   To quantify a token's contribution to the *target* distribution, we compare predictions at $T$ with and without information from $x_i$. For each token $x_i$, we mask it and measure how the output distribution over the target token changes. For a chosen scheme, let $P_{\text{orig}}(\cdot \mid x_{<T})$ and $P_{\text{masked}}^{(i)}(\cdot \mid x_{<T})$ denote the target distributions. The information contribution is

$$\mathcal{I}(x_i \to x_T) \;=\; D_{\text{KL}}\Big( P_{\text{orig}}(\cdot \mid x_{<T}) \,\big\|\, P_{\text{masked}}^{(i)}(\cdot \mid x_{<T}) \Big). \tag{4}$$

**Final Attribution Score**   We combine the three components into a target-conditioned attribution:

$$\text{Attr}(x_i \to x_T) \;=\; M_T[i] \left( \beta\, S_i^{(T)} \;+\; \gamma\, \mathcal{I}(x_i \to x_T) \right), \tag{5}$$

where $\beta, \gamma \geq 0$ weight curvature-based sensitivity and information contribution, respectively. The gate $M_T[i]$ restricts attribution to tokens with causal paths to the target and redistributes mass over such paths. In conjunction with the scalable HVP-based curvature estimator ( equation 3) and the output-aware information term ( equation 4),  equation 5 yields a causally grounded, curvature-aware, and robust token-level attribution tailored for decoder-only generative models.

An overview of our method is illustrated in Figure 1, with the full algorithmic procedure provided in Appendix A3. The theoretical properties and error bounds of HETA are discussed in detail in Appendix A2.

## 5 Experiments and Results

### 5.1 Experimental Setup and Datasets

To evaluate the proposed HETA framework, we conduct experiments on both benchmark and curated datasets covering a wide range of reasoning and generation complexity.

We use established benchmarks from Zhao & Shan (2024). The datasets considered are: (1) **Long-Range Agreement (LongRA)** Vafa et al. (2021), which evaluates a model's ability to maintain coherence across long-distance semantic dependencies by inserting distractor sentences between related word pairs (e.g., "Japan" and "Tokyo"); (2) **TellMeWhy** Lal et al. (2021), a narrative QA dataset that requires multi-sentence causal reasoning to explain a character's motivations; and (3) **WikiBio** Manakul et al. (2023), composed of structured Wikipedia biographies where the task involves generating plausible and factual sentence continuations from short prompts.In addition, we introduce

a carefully **curated dataset** of 2,000 mixed-paragraph instances to evaluate whether attribution aligns with the truly predictive evidence in context. Each instance concatenates one narrative segment from NARRATIVEQA Kočiskỳ et al. (2018) with the answer-bearing support segment from SCIQ Johannes Welbl (2017), followed by the corresponding SCIQ question; the model then generates the first answer token, and attributions are computed with respect to this onset token so the model has access to the full paragraph and question before attribution is measured. For example:

> *The protagonist returns to the village after the winter storm, reflecting on her father's passing. Photosynthesis primarily occurs in the leaves of the plant, where chloroplasts capture light. Question: In which part of the plant does photosynthesis mainly take place?*

Here, the correct target is *leaves*, and the meaningful contributing tokens lie in the second (SciQ) segment; the first (NarrativeQA) segment is semantically rich but non-diagnostic for the question. Answer spans and minimal supporting cues in the SciQ segment are automatically annotated by two independent systems (GPT-4o and GPT-5 Thinking), and we take their *intersection* at the subword level to form high-precision labels; inter-annotator agreement is high (**$F_1$ = 0.91**, **Cohen's $\kappa$ = 0.89**) across the corpus. To quantify alignment, we report the *Dependent Sentence Attribution* score (described below), which contrasts attribution mass on annotated tokens in the SciQ segment against mass placed on the entire NarrativeQA segment after per-instance normalization. This construction provides a compact, interpretable probe of target-conditioned attribution quality.

HETA is compared against a broad suite of attribution methods: **ContextCite**Cohen-Wang et al. (2024), **Integrated Gradients**Sundararajan et al. (2017), **Peering into the Mind of LMs (PML)**Phukan et al. (2024), **TDD-backward**Feng et al. (2024), **attention rollout**Abnar & Zuidema (2020), **fAML**Barkan et al. (2024), **Progressive Inference**Kariyappa et al. (2024), **SEA-CoT**Palikhe et al. (2025), and **ReAgent**Zhao & Shan (2024). To measure attribution faithfulness, we use **Soft-NC** and **Soft-NS** Zhao & Aletras (2023), modified for generative models as in Zhao & Shan (2024), which assess how output distributions shift under input perturbation based on attribution scores.

Table 1: **Attribution alignment on the curated dataset** using the (Dependent Sentence Attribution) metric. We evaluate robustness across both model size and architecture by testing HETA on diverse decoder-only LMs, spanning different parameter scales and design choices. Higher scores indicate stronger alignment with human-annotated tokens. HETA outperforms all baselines.

| Method | GPT-J | LLaMA | Phi-3 | Qwen2.5 |
|---|---|---|---|---|
| IG | -0.34 | -0.28 | -0.41 | -0.31 |
| ContextCite | -0.12 | -0.09 | -0.18 | -0.14 |
| Peering (PML) | -0.25 | -0.21 | -0.30 | -0.22 |
| TDD-backward | -0.31 | -0.27 | -0.36 | -0.29 |
| Attention Rollout | -0.44 | -0.39 | -0.52 | -0.41 |
| fAML | 2.10 | 2.30 | 2.05 | 2.20 |
| Progressive Inference | 2.65 | 2.88 | 2.40 | 2.73 |
| SEA-CoT | 2.92 | 3.15 | 2.77 | 2.85 |
| ReAgent | 3.60 | 3.78 | 3.35 | 3.50 |
| **HETA (Ours)** | **4.80** | **5.10** | **4.25** | **4.65** |

**Controlled Attribution Evaluation via the DSA Metric.** To assess attribution alignment on the curated dataset, we introduce the **Dependent Sentence Attribution (DSA)** metric. This metric quantifies the degree to which attribution mass is correctly concentrated on the answer-relevant portion of the input, specifically, the second part of each curated paragraph (SciQ), which contains the evidence required to answer the question.

Formally, let $S_{NarrQA}$ and $S_{SciQ}$ denote the set of model-selected important token indices within the first and second part of the curated text, and let $ss_i$ and $fs_i$ be the normalized attribution scores assigned to token $i$ in the second and first part, respectively; the DSA score is defined as DSA $= \sum_{i \in S_{SciQ}} ss_i - \sum_{j \in S_{NarrQA}} fs_j$, with attributions normalized so the total mass over the paragraph sums to one and the final DSA reported as the average over all instances.

Higher DSA values indicate that the attribution method assigns more mass to the truly predictive evidence (in the second part) and less to unrelated context (in the first part), thereby reflecting better alignment with causal semantics. DSA complements traditional faithfulness metrics by directly evaluating attribution precision under a controlled, interpretable input structure.

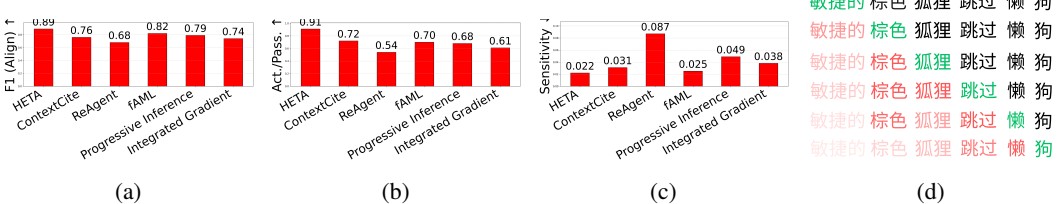

Figure 2: **(a)-(c) Analysis of HETA components.** Each bar plot shows the effect of ablating key components of HETA. The full HETA model achieves the highest attribution faithfulness and alignment across all metrics, while removing individual components consistently degrades performance.(d) Input importance distributions for a generative task using our proposed HETA method.

Figure 3: **(a)-(c) Analysis of robustness of HETA vs baseline methods (Left)** Sensitivity under Gaussian perturbations (lower is better), where HETA maintains the lowest variance across input noise. **(Center)** Active/Passive robustness (higher is better), reflecting attribution consistency across syntactic rephrasings. **(Right)** Alignment F1 score against annotated tokens (higher is better). HETA outperforms all baselines, validating the complementary role of transition flow, curvature, and information gain.(d) Input importance distributions for a generative task using our proposed HETA method using Qwen 2.5 3B.

We evaluate attribution quality using both perturbation-based faithfulness metrics (Soft-NC, Soft-NS) and alignment-based analysis on a curated dataset (DSA). Experiments are conducted across four transformer models: **Qwen2.5-3B (Alibaba Qwen)**[2](results are shown in AppendixA3 due to space constraint), **GPT-J-6B (EleutherAI)Wang & Komatsuzaki (2021):** [3], **Phi-3-Medium-4K-Instruct (14B, Microsoft):** [4], and **Llama-3.1-70B (Meta)** [5]. This selection enables analysis across varying model capacities, architectures, and parameter scales. For the Qwen model the results are shown in Appendix A3 due to space constraints.

**Hyperparameters and compute.** All experiments run on a single NVIDIA A100 (80 GB). Unless noted, we set the HETA weights in the final score to $\beta = 0.5$ (Hessian sensitivity) and $\gamma = 0.5$ (KL information) and evaluate sequences of length 512 with batch size 16; long-context runs use length 2,048 with windowing and batch size 8. For efficiency, our default is a **low-rank Hessian with windowing (HETA-LR+WIN)**: a rank-64 blockwise low-rank estimator applied within 512-token windows with 50% overlap. This variant closely matches full HETA while substantially reducing cost (see Appendix Tables 8, 9). For **LLaMA-3.1-70B**, we further cut compute by computing second-order terms only in the *last six layers*, which preserves attribution quality in practice. For smaller models (e.g., GPT-J, OPT), we compute second-order terms across *all layers* unless otherwise specified. In ablations, we also report **HETA-LR** (rank = 64) without windowing and **HETA-LS** (6-layer sampling) without low-rank, keeping all other hyperparameters fixed for a fair comparison.

## 5.2 RESULTS

According to Table 3, across all model-task combinations, **HETA achieves the highest Soft-NC and Soft-NS scores**, demonstrating superior attribution robustness under input perturbations. For instance,

---

[2]https://huggingface.co/Qwen/Qwen2.5-3B

[3]https://huggingface.co/EleutherAI/gpt-j-6b

[4]https://huggingface.co/microsoft/Phi-3-medium-4k-instruct

[5]https://huggingface.co/meta-llama/Llama-3.1-70B

Table 2: **Sensitivity of attribution metrics to decoding hyperparameters** (max relative change $\Delta\%$; lower is better). HETA varies $< 1\%$ across all models/metrics; each baseline fluctuates $> 2\%$. Grid: temperature $\in \{0.2, 0.5, 0.9\}$, top-$p \in \{0.8, 0.9, 0.95\}$, top-$k \in \{20, 50, 100\}$, repetition penalty $\in \{1.0, 1.2\}$; 3 seeds.

| Model | Metric | HETA | ContextCite | IG | PML | TDD-bw | AttnRoll | fAML | ProgInf | SEA-CoT | ReAgent |
|---|---|---|---|---|---|---|---|---|---|---|---|
| | Soft-NC | **0.6** | 2.6 | 2.7 | 2.4 | 3.1 | 2.3 | 2.5 | 2.4 | 2.5 | 2.3 |
| Llama-3.1 70B | Soft-NS | **0.8** | 3.1 | 3.0 | 2.8 | 3.4 | 2.7 | 2.9 | 2.8 | 3.0 | 2.6 |
| | DSA | **0.6** | 2.4 | 2.3 | 2.2 | 2.8 | 2.1 | 2.3 | 2.3 | 2.4 | 2.2 |

on GPT-J 6B, HETA attains a Soft-NC of 10.3 on LONGRA and 9.2 on TELLMEWHY—exceeding the next best method, ReAgent, by over $2\times$. Similar trends hold across Phi-3 and Llama-3.1, confirming HETA's effectiveness across model scales. While **ReAgent consistently ranks second**, recent methods such as SEA-CoT, and Progressive Inference show moderate improvements over traditional techniques. In contrast, *Integrated Gradients* and attention-based variants often yield low or negative Soft-NS values, indicating instability and low attribution faithfulness.

To complement the above, we assess attribution alignment using the DSA metric on a curated dataset (Table 1). Again, **HETA outperforms all baselines by a substantial margin**, achieving DSA scores $\geq 4.2$ across all models. **ReAgent remains the strongest non-HETA method**, followed by SEA-CoT. In contrast, gradient- and attention-based methods yield negative DSA values, highlighting their inability to isolate causal tokens in the presence of distractors. **Results are averaged across all datasets with GPT-J-6B; higher is better. Mean performance is reported over three independent runs with standard deviation** $< 0.2$. These results collectively indicate that **HETA provides both faithful and semantically aligned attributions**, setting a new state-of-the-art across both benchmark and controlled evaluation settings (please see Appendix A3 for further details). We illustrate HETA's token-level attributions with two qualitative examples in Figures 2(d) and 3(d).

## 5.3 ROBUSTNESS OF HETA

To further demonstrate the robustness of our methodology, we performed a stress test using three complementary attribution metrics: *Sensitivity*, *Active/Passive Robustness*, and *F1 (Alignment)*. We use the Phi-3 medium model and the TellMeWhy dataset for stress test and ablation studies unless otherwise mentioned. These were evaluated across the six ablated configurations: **(1) Sensitivity** quantifies attribution stability under small perturbations. For each token embedding $X_i$, we add Gaussian noise $\epsilon \sim \mathcal{N}(0, \delta^2 I)$, compute attribution scores over multiple perturbations, and report the average per-token standard deviation: Sensitivity $= \frac{1}{T} \sum_{i=1}^{T} \sigma_i$, where $\sigma_i$ denotes the standard deviation of the attribution score for token $i$, and $T$ is the sequence length. **(2) Active/Passive Robustness** measures syntactic invariance. Given an original sentence and its active/passive rephrasing, we align corresponding tokens and compute the Spearman rank correlation between their attribution rankings: Robustness $= \rho(\text{Attr}(x_i \rightarrow x_T), \text{Attr}(x_i' \rightarrow x_T'))$. **(3) F1 (Alignment)** evaluates agreement between model attributions and annotations made by GPT-4o and GPT5 in our curated dataset. Let $\mathcal{A}_{\text{model}}$ denote the top-attributed tokens and $\mathcal{A}_{\text{anno.}}$ the gold-annotated set. The F1 score is computed as F1 $= \frac{2 |\mathcal{A}_{\text{model}} \cap \mathcal{A}_{\text{anno.}}|}{|\mathcal{A}_{\text{model}}| + |\mathcal{A}_{\text{anno.}}|}$.

Figure 3 (a-c) shows that the HETA framework consistently yields the lowest sensitivity and the highest robustness and F1 scores compared to the baseline methods, indicating stable, syntax-invariant, and logically-aligned attributions, emphasizing the complementary roles of semantic flow, curvature information, and information gain in delivering reliable token-level attribution.

## 5.4 ROBUSTNESS TO DECODING HYPERPARAMETERS

We evaluate the sensitivity of attribution quality to common decoding hyperparameters. For each method and model, we sweep a fixed grid, temperature $\{0.2, 0.5, 0.9\}$, top-$p$ $\{0.8, 0.9, 0.95\}$, top-$k$ $\{20, 50, 100\}$, and repetition penalty $\{1.0, 1.2\}$, across three random seeds. For each metric, we report the *maximum relative change* $\Delta\%$ across the grid (lower is better).

Table 2 shows that *HETA's* attribution metrics remain effectively invariant to decoding settings, with worst-case $\Delta\% < 1$ for Soft-NC, Soft-NS, and DSA (see AppendixA4 for more details). In contrast,

| Attribution Method | LongRA | | TellMeWhy | | WikiBio | |
|---|---|---|---|---|---|---|
| | Soft-NC↑ | Soft-NS↑ | Soft-NC↑ | Soft-NS↑ | Soft-NC↑ | Soft-NS↑ |
| **GPT-J 6B** | | | | | | |
| ContextCite | 1.42 | 0.03 | 1.46 | -0.22 | 0.49 | -0.08 |
| Integrated Gradients | 1.87 | 0.45 | 1.54 | 0.04 | 1.38 | 0.77 |
| Peering (PML) | 2.05 | 0.50 | 1.68 | 0.06 | 1.50 | 0.83 |
| TDD-backward | 1.10 | -0.12 | 1.89 | -0.03 | 0.11 | 0.51 |
| Attention Rollout | 0.41 | -0.01 | 0.25 | -0.09 | 1.91 | 0.46 |
| fAML | 0.21 | -0.10 | 0.05 | -0.09 | 0.21 | -0.02 |
| Progressive Inference | 1.35 | 0.28 | 1.12 | 0.25 | 0.99 | 0.22 |
| SEA-CoT | 1.54 | 0.32 | 1.30 | 0.31 | 1.10 | 0.35 |
| ReAgent | 1.68 | 0.37 | 1.45 | 0.36 | 1.22 | 0.39 |
| **HETA (Ours)** | **10.3** | **2.31** | **9.2** | **2.04** | **3.80** | **2.20** |
| **Phi-3-Medium-14B** | | | | | | |
| ContextCite | 1.50 | 0.04 | 1.45 | -0.20 | 0.52 | -0.06 |
| Integrated Gradients | 1.95 | 0.44 | 1.60 | 0.06 | 1.35 | 0.70 |
| Peering (PML) | 2.15 | 0.49 | 1.75 | 0.08 | 1.48 | 0.76 |
| TDD-backward | 1.05 | -0.10 | 1.82 | -0.02 | 0.10 | 0.50 |
| Attention Rollout | 0.39 | -0.02 | 0.30 | -0.08 | 1.85 | 0.43 |
| fAML | 0.23 | -0.09 | 0.08 | -0.10 | 0.20 | -0.04 |
| Progressive Inference | 1.30 | 0.25 | 1.18 | 0.26 | 1.00 | 0.21 |
| SEA-CoT | 1.50 | 0.31 | 1.32 | 0.33 | 1.15 | 0.34 |
| ReAgent | 1.66 | 0.38 | 1.47 | 0.39 | 1.25 | 0.40 |
| **HETA (Ours)** | **10.8** | **2.35** | **9.5** | **2.20** | **4.20** | **2.30** |
| **LLaMA-3.1 70B** | | | | | | |
| ContextCite | 1.17 | 0.58 | 1.20 | 0.56 | 0.85 | 0.57 |
| Integrated Gradients | 0.13 | 0.13 | 0.13 | 0.10 | 0.13 | 1.15 |
| Peering (PML) | 0.15 | 0.15 | 0.15 | 0.12 | 0.15 | 1.20 |
| TDD-backward | -0.02 | -0.11 | 0.01 | -0.10 | -0.02 | 0.59 |
| Attention Rollout | -1.48 | 0.01 | -1.48 | 0.01 | -1.48 | 0.61 |
| fAML | 0.46 | -0.21 | 0.46 | -0.21 | 0.46 | -0.07 |
| Progressive Inference | 1.20 | 0.24 | 1.00 | 0.22 | 0.95 | 0.26 |
| SEA-CoT | 1.35 | 0.30 | 1.15 | 0.28 | 1.05 | 0.31 |
| ReAgent | 1.55 | 0.36 | 1.28 | 0.34 | 1.15 | 0.38 |
| **HETA (Ours)** | **9.9** | **2.60** | **8.6** | **2.25** | **3.70** | **2.10** |

Table 3: **Attribution faithfulness on benchmark datasets** (LongRA, TellMeWhy, WikiBio) using GPT-J 6B, Phi-3-Medium-14B, and LLaMA-3.1 70B. Evaluated with Soft-NC and Soft-NS; higher is better. Mean of 3 runs; std $< \pm0.06$.

all baselines exhibit substantially larger variability, typically 2–5%. HETA's stability arises from three design elements: a target-conditioned causal gate that confines credit to paths terminating at the current prediction, a curvature-aware sensitivity term that smooths local logit perturbations, and an information-theoretic component that scores distributional shifts rather than single sampled outcomes. These jointly decouple attribution from stochastic decoding heuristics (temperature, top-$p$, top-$k$), whereas ablation-, gradient-, and similarity-based baselines depend more directly on sampled logits or linear approximations and thus vary markedly with hyperparameter changes.

## 6 ABLATION STUDIES

To assess the contribution of each component in **HETA**, we conduct a comprehensive ablation study in this section. Due to space constraints, a detailed ablation study is provided in the Appendix A4. Experiments are performed using the **GPT-J 6B** model on three benchmark datasets, **LongRA**, **TellMeWhy**, and **WikiBio**, along with the curated attribution dataset introduced in Section 5. We compare six configurations: (1) the full HETA model (Transition + Hessian + KL), (2) Transition

Only, (3) Hessian Only, (4) KL Only, (5) No Transition Gating (Hessian + KL without semantic weighting), and (6) Uniform Transition (equal token weighting instead of the learned semantic transition vector $M_T$). Performance is evaluated using the same metrics as our main experiments: **Soft-NC** and **Soft-NS** for attribution sensitivity on benchmark datasets, and **DSA** for alignment with human-annotated tokens on the curated dataset. We set the aggregation hyperparameters to $\beta = 0.5$ and $\gamma = 0.5$, and compute KL divergence using masked-token perturbation. All reported results are averaged across 1000 randomly sampled instances per dataset. Results in Figure 2(a–c) demonstrate that each component contributes meaningfully to HETA's performance. Removing the semantic transition vector ($M_T$) or replacing it with uniform weighting leads to significant drops in all metrics, confirming the importance of modeling directional semantic influence across layers. Similarly, Hessian-based sensitivity and KL-based information measures provide complementary improvements by capturing curvature-sensitive effects and token-level information contributions

## 7 RELATED WORKS

Global explainability methods aim to extract broader patterns from LLMs. Probing techniques have been instrumental in identifying syntactic and semantic representations encoded in LLMs Hewitt & Manning (2019); Peng et al. (2022). Studies by Geva et al. (2022) and Kobayashi et al. (2023) show that feed-forward layers and attention heads capture complex linguistic knowledge. Mechanistic interpretability, as explored by Wang et al. (2022), seeks to reverse-engineer neural networks into comprehensible circuits, facilitating a deeper understanding of tasks like object identification. Model editing techniques have also emerged as a promising area for explainability. Hypernetwork-based editing Mitchell et al. (2022) and causal tracing Meng et al. (2022) enable targeted modifications in model behavior without extensive retraining, allowing models to adapt to specific inputs while maintaining overall performance Yao et al. (2023).

Local feature attribution methods such as LIME Ribeiro et al. (2016), KernelSHAP Lundberg & Lee (2017), Integrated Gradients Sundararajan et al. (2017), Grad-CAM Selvaraju et al. (2017), and LRP Bach et al. (2015) quantify per-feature contributions to a model's output. However, most rely on first-order or linear approximations that often produce inconsistent attributions Hooker et al. (2019) and break down in the nonlinear, contextual regimes typical of autoregressive generation. Attention-based interpretability Abnar & Zuidema (2020) is similarly limited: attention weights reflect where a model looks rather than what causally influences its output, and can be perturbed without significantly altering predictions Jain & Wallace (2019).

Several recent methods target attribution in generative LLMs specifically. ContextCite Cohen-Wang et al. (2024) trains a sparse linear surrogate via ablations but is limited to sentence-level attribution and sensitive to redundancy. TDD Feng et al. (2024) projects hidden states through the output head (logit lens), conflating correlation with causation. *Peering into the Mind of LMs* Phukan et al. (2024) matches hidden-state representations via cosine similarity, performing well on verbatim spans but lacking robustness to paraphrasing or indirect evidence. ReAGent Zhao & Shan (2024) offers a model-agnostic approach for generative settings but ignores dense semantic structure in internal layers Chen et al. (2024), limiting its ability to capture deep token-level influence. These limitations motivate the need for attribution frameworks that combine causal structure, higher-order sensitivity, and information-theoretic grounding.

## 8 CONCLUSION AND LIMITATIONS

We introduced HETA, a unified framework that improves attribution faithfulness and robustness over strong baselines by combining causal semantic gating, Hessian-based curvature, and KL-divergence-based information gain into a principled token-level attribution mechanism for decoder-only language models. However, it incurs higher runtime, greater memory usage, and reduced efficiency on long texts. These trade-offs highlight the need for optimization, and future work will explore low-rank approximations and layer sampling for better scalability (see Appendix A5 for details).

## ACKNOWLEDGMENT

This work was supported in part by the National Science Foundation under Grant No. 2404036, by University of Florida startup funds, and by the Defense Advanced Research Projects Agency (DARPA) under Contracts No. HR00112490420 and No. HR00112420004. Any opinions, findings, conclusions, or recommendations expressed in this material are those of the authors and do not necessarily reflect the views of the sponsoring agencies.

## ETHICS STATEMENT

We affirm adherence to the ICLR Code of Ethics and have designed this work to minimize risks related to privacy, safety, and misuse. Our experiments use publicly available datasets (NARRATIVEQA and SCIQ) under their respective licenses; no personally identifiable information or sensitive user data are collected, and no human subjects research requiring IRB oversight was conducted. The curated evaluation set is constructed from these sources and automatically annotated by two independent LMs (GPT-4o and GPT-5-Thinking) with intersection-based labels to reduce over-selection; we report inter-annotator agreement and provide full documentation of preprocessing, tokenization, and filtering to support reproducibility. While attribution methods can be misused for model extraction or targeted prompt manipulation, we mitigate these risks by reporting aggregate metrics, withholding raw internal activations, and releasing code with rate limits and intended-use guidelines. We discuss known limitations—e.g., approximation error under low-rank/windowed Hessians and the use of attention-value rollout as a causal gate—and provide diagnostics and ablations to avoid overstating causal claims. All compute was performed on institutional infrastructure (A100 80GB class GPUs); we report hardware footprints and favor efficient approximations to reduce energy use. We disclose that the authors have no conflicts of interest or external sponsorship that bias this study. Upon publication, we plan to release code and scripts under a permissive license with documentation on safe and responsible use.

## REPRODUCIBILITY STATEMENT

We take reproducibility seriously and provide all necessary artifacts to re-create our results. The paper specifies the full method in Sections 1–5 and Algorithm 1; theoretical assumptions, bounds, and proofs (including low-rank/windowed error guarantees) are in Appendix A2. Datasets and preprocessing are documented in Section A3.1 (NarrativeQA⊕SciQ construction, prompts for automatic annotation, fixed answer-support intersection labels with agreement scores) together with release scripts that regenerate the curated set from the original sources.

**Supplementary materials.** We include the full HETA algorithm listing (Algorithm 1) and submit our complete, anonymized codebase as supplementary material, containing: (i) reference implementations of HETA and its LR+WIN approximation (deterministic seeds, PyTorch/`transformers` versions, environment file), (ii) end-to-end evaluation pipelines for all reported models (GPT-J 6B, LLaMA-3.1 70B, Phi-3 14B, Qwen2.5 3B), (iii) scripts to reproduce every table/figure and compute all metrics, and (iv) prompts/configs used for automatic token-level supervision. The repository includes a `RUN.md` with single-command entry points and fixed random seeds to match table numbers within reported standard deviations.

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

APPENDIX

CONTENTS

## A1 WHY GRADIENTS AND INTEGRATED GRADIENTS CAN BOTH FAIL IN FLAT REGIONS

In this section, we expand on the toy example from Section 3.1 to illustrate, in detail and without ambiguity, how both standard gradient-based attribution and Integrated Gradients (IG) can assign *zero* (or negligible) importance to an input feature even when that feature clearly influences the model's output in a nearby region. The phenomenon arises in networks with piecewise-linear activations such as ReLU, which create locally flat regions where pointwise gradients vanish; a closely related "near-flat" effect occurs for smooth, saturating nonlinearities (e.g., GELU/softplus), where gradients along a chosen baseline-to-input path can be uniformly tiny.

### A1.1 GRADIENT FAILURE IN ReLU FLAT REGIONS

Consider

$$f(x) = \mathrm{ReLU}(w^\top x + b), \qquad x = \begin{bmatrix} x_1 \\ x_2 \end{bmatrix} \in \mathbb{R}^2, \quad w = \begin{bmatrix} 1 \\ 1 \end{bmatrix}, \quad b = -2.$$

Take the input

$$x_0 = \begin{bmatrix} 0 \\ 0 \end{bmatrix}.$$

Then

$$w^\top x_0 + b = 0 + 0 - 2 = -2 < 0 \ \Rightarrow \ f(x_0) = \mathrm{ReLU}(-2) = 0,$$

so $x_0$ lies in a flat (inactive) region of the ReLU. The derivative of ReLU is

$$\frac{d}{dz}\mathrm{ReLU}(z) = \begin{cases} 1, & z > 0, \\ 0, & z < 0, \\ \text{undefined (often set to 0)}, & z = 0, \end{cases}$$

and by the chain rule,

$$\nabla f(x_0) = \frac{d}{dz}\mathrm{ReLU}(w^\top x_0 + b) \, \nabla_x(w^\top x)\Big|_{x=x_0} = 0 \cdot w = \begin{bmatrix} 0 \\ 0 \end{bmatrix}.$$

**Implication.** Any *pointwise* gradient-based attribution at $x_0$ is identically zero, even though a small displacement can cross the hinge and change the output. For instance, with

$$x = \begin{bmatrix} 2.1 \\ 0 \end{bmatrix}, \qquad w^\top x + b = 2.1 - 2 = 0.1 > 0 \ \Rightarrow \ f(x) = 0.1 \neq 0.$$

Thus, zero gradient at $x_0$ does not imply the feature is globally unimportant; it reflects the local flatness of the activation at that point.

### A1.2 INTEGRATED GRADIENTS FAILURE ALONG FLAT (OR NEAR-FLAT) PATHS

Integrated Gradients (IG) aims to mitigate pointwise gradient pathologies by integrating gradients along a path from a baseline $x'$ to the input $x$:

$$\mathrm{IG}_i(x; x') \ = \ (x_i - x'_i) \int_{\alpha=0}^{1} \frac{\partial f(x' + \alpha(x - x'))}{\partial x_i} \, d\alpha.$$

However, IG still depends on the *gradients along the chosen path*. If that path remains entirely within a flat region (or within a region where the pre-activation stays negative), every integrand vanishes and the IG attribution becomes zero.

**Exact flat-path example (ReLU).** Use the same $f$ and choose the baseline $x' = \begin{bmatrix} -1 \\ -1 \end{bmatrix}$ and the input $x_0 = \begin{bmatrix} 0 \\ 0 \end{bmatrix}$. The straight-line path is

$$x(\alpha) = x' + \alpha(x_0 - x') = \begin{bmatrix} -1 + \alpha \\ -1 + \alpha \end{bmatrix}, \qquad \alpha \in [0, 1].$$

Along this path,

$$w^\top x(\alpha) + b = (-1 + \alpha) + (-1 + \alpha) - 2 = -4 + 2\alpha < 0 \quad \text{for all } \alpha \in [0, 1],$$

so $f(x(\alpha)) = 0$ and $\nabla f(x(\alpha)) = \mathbf{0}$ everywhere on the path. Therefore,

$$\text{IG}(x_0; x') = \mathbf{0}.$$

Yet, as shown above, moving slightly away from $x_0$ (e.g., to $x = [2.1, 0]^\top$) increases the output, indicating that the inputs *do* influence $f$ beyond the flat region.

**Baseline/path dependence (why IG can succeed or fail).** IG is *baseline-dependent*. If we chose instead an endpoint $x$ whose straight path from $x'$ crosses the hinge (i.e., for some $\alpha^\star$, $w^\top x(\alpha^\star) + b = 0$), then gradients on a nontrivial subinterval would be nonzero and IG would assign positive attribution. In the $x_0$ case above, IG vanishes because the chosen path lies entirely in the inactive region. Thus, IG can fail at inputs that lie inside flat regions *for certain baselines/paths*, even though the function responds immediately outside that region.

**Near-flat smooth activations (GELU/softplus).** For smooth saturating nonlinearities (e.g., $\text{softplus}(z) = \log(1 + e^z)$ or GELU), strict flatness is replaced by *very small* gradients when $z \ll 0$. If the baseline-to-input path remains in a "near-flat" zone where $z(\alpha) = w^\top x(\alpha) + b \ll 0$ for all $\alpha \in [0, 1]$, then

$$\|\nabla f(x(\alpha))\| \approx 0 \quad \forall \alpha,$$

and IG can be arbitrarily small, under-attributing features despite non-negligible finite changes for steps that push $z$ toward the transition region. Hence, the qualitative failure mode persists in smooth networks: the path integral can be dominated by a region of tiny gradients, producing near-zero IG.

### A1.3 BROADER IMPLICATIONS

These effects extend beyond toy settings. In transformer LMs, *(i)* causal masking and attention patterns can route influence away from certain tokens along specific layers/paths; *(ii)* residual/MLP mixing plus saturating activations can create local neighborhoods where pointwise gradients (and path-integrated gradients for common baselines) are nearly zero; yet *(iii)* modest, structured perturbations to those tokens can meaningfully alter the output distribution at a downstream position. Consequently, both raw gradients and IG may *under-attribute* token importance in regions of flat or near-flat sensitivity.

**Takeaway and motivation.** Pointwise gradients fail at flat inputs; IG can also fail when its baseline-to-input path lies in (near-)flat regions or skirts the true transition surface that mediates the output change. These limitations motivate complementary signals: curvature-aware sensitivity (Hessian-based), target-conditioned semantic flow that respects causal routing, and information-theoretic change (e.g., KL under masking). Together, they capture influence that first-order and path-integrated gradients can miss.

## A2 THEORETICAL FOUNDATIONS AND PROPERTIES OF HETA

In this section, we provide a mathematically rigorous foundation for the Hessian-Enhanced Token Attribution (HETA) framework. We formalize attribution as a decomposition problem for the target log-likelihood, establish faithfulness error bounds, and show how combining semantic flow, Hessian curvature, and KL-based information contributions can improve faithfulness and robustness relative to single-view attribution methods.

### A2.1 PRELIMINARIES

Consider a decoder-only language model $f_\theta$ with parameters $\theta$, input tokens $x_{1:T}$, embeddings $\mathbf{X} \in \mathbb{R}^{T \times d}$, and the target conditional distribution

$$P_\theta(x_T \mid x_{<T}) = \text{Softmax}\big(f_\theta(\mathbf{X})\big).$$

Define the target log-likelihood

$$g(\mathbf{X}) \;=\; \log P_\theta(x_T \mid x_{<T}).$$

Let $\mathbf{X}_{\setminus R}$ denote the embeddings where a subset $R \subseteq \{1, \ldots, T-1\}$ of *context* tokens is replaced by a masking scheme (e.g., zeroed embedding, mean embedding, or a learned sentinel used only at evaluation time). An attribution method produces scores $\mathrm{Attr}(x_i) \geq 0$ such that

$$\sum_{i=1}^{T-1} \mathrm{Attr}(x_i) \;\approx\; g(\mathbf{X}) \;-\; g(\mathbf{X}_{\text{all masked}}),$$

where $\mathbf{X}_{\text{all masked}}$ masks all context tokens $\{1, \ldots, T-1\}$. We measure *faithfulness error* on a subset $R$ by

$$\mathcal{L}(\mathrm{Attr}) \;=\; \Big| g(\mathbf{X}) - g(\mathbf{X}_{\setminus R}) \;-\; \sum_{i \in R} \mathrm{Attr}(x_i) \Big|,$$

so smaller $\mathcal{L}(\mathrm{Attr})$ indicates more faithful attribution.

**Divergence-Based Attribution Lower Bound.** HETA uses Kullback–Leibler divergence to quantify the information contribution of each token $x_i$ to the target $x_T$. Let $P_{\text{orig}}(\cdot) = P_\theta(\cdot \mid x_{<T})$ and let $P^{(i)}_{\text{masked}}(\cdot)$ be the target distribution when only $x_i$ (with $i < T$) is masked according to a chosen scheme while all other context tokens are kept. Define the total-variation distance

$$\delta_i \;:=\; \big\| P_{\text{orig}} - P^{(i)}_{\text{masked}} \big\|_1.$$

By Pinsker's inequality (with natural logarithms),

$$D_{\mathrm{KL}}\big(P_{\text{orig}} \,\big\|\, P^{(i)}_{\text{masked}}\big) \;\geq\; \tfrac{1}{2}\, \delta_i^2.$$

With HETA's final form

$$\mathrm{Attr}(x_i \to x_T) \;=\; M_T[i]\, \Big( \beta\, S_i^{(T)} \;+\; \gamma\, D_{\mathrm{KL}}\big(P_{\text{orig}} \,\big\|\, P^{(i)}_{\text{masked}}\big) \Big),$$

it follows that

$$\mathrm{Attr}(x_i \to x_T) \;\geq\; M_T[i]\, \gamma\, \tfrac{1}{2}\, \delta_i^2.$$

Thus any token whose masking substantially perturbs the target distribution receives nontrivial attribution, modulated by its target-conditioned transition mass $M_T[i]$.

**Norm Bounds on Hessian Sensitivity.** To model second-order curvature, HETA considers the Hessian

$$H_T \;=\; \nabla^2_{\mathbf{X}} \log P_\theta(x_T \mid x_{<T}) \;\in\; \mathbb{R}^{(Td) \times (Td)},$$

where $\mathbf{X} \in \mathbb{R}^{T \times d}$ are the input embeddings. The sensitivity of token $x_i$ is the entrywise $\ell_1$-mass of the Hessian rows corresponding to its $d$-dimensional block:

$$S_i^{(T)} \;=\; \sum_{j=1}^{Td} \Big| H_T[i \cdot d : (i+1) \cdot d,\, j] \Big| \;=\; \big\| \Pi_i H_T \big\|_{1,1},$$

with $\Pi_i$ selecting the block rows for token $i$. By standard norm relations,

$$S_i^{(T)} \;\leq\; \|H_T\|_{1,1} \;\leq\; (Td)\, \|H_T\|_F,$$

where $\|\cdot\|_{1,1}$ is the entrywise $\ell_1$ norm and $\|\cdot\|_F$ the Frobenius norm (the last inequality uses $\|A\|_{1,1} \leq \sqrt{N}\|A\|_F$ with $N = (Td)^2$). These bounds control each token's curvature-based sensitivity by the global curvature magnitude measured in Frobenius/entrywise norms.

**Attribution Envelope from Information Loss.** Let the (signed) log-probability change from masking $x_i$ be

$$\Delta_i \;:=\; \log P_\theta(x_T \mid x_{<T}) \;-\; \log P_\theta\big(x_T \mid x_{<T \setminus \{x_i\}}\big),$$

and define its positive part $\Delta_i^+ = \max\{0, \Delta_i\}$. HETA's final score (cf. equation 5) uses the KL term with weight $\gamma$:

$$\mathrm{Attr}(x_i \to x_T) \;=\; M_T[i]\Big( \beta\, S_i^{(T)} + \gamma\, D_{\mathrm{KL}}\big(P_{\text{orig}} \,\big\|\, P^{(i)}_{\text{masked}}\big)\Big).$$

For interpretability we report the envelope

$$\text{Env}(x_i \to x_T) \;=\; M_T[i]\big(\beta\, S_i^{(T)} + \gamma\, \Delta_i^+\big),$$

which upper-bounds the final score whenever $D_{\text{KL}}\big(P_{\text{orig}} \,\|\, P_{\text{masked}}^{(i)}\big) \leq \Delta_i^+$ (a condition we verify empirically for our masking schemes). This provides a conservative, target-linked scale for attribution magnitudes without assuming monotonicity of log-probability under masking.

**Functional Faithfulness via a Taylor Remainder.** Let $g(\mathbf{X}) = \log P_\theta(x_T \mid x_{<T})$. For a perturbation $\epsilon_i \in \mathbb{R}^d$ applied to token $i$'s embedding block, a second-order expansion yields

$$g(\mathbf{X} + \epsilon_i) \;\approx\; g(\mathbf{X}) \;+\; \langle \nabla_{x_i} g, \epsilon_i \rangle \;+\; \tfrac{1}{2}\epsilon_i^\top H_{x_i x_i}\,\epsilon_i,$$

where $H_{x_i x_i} \in \mathbb{R}^{d \times d}$ is the token-specific block of $H_T$. If $g$ is $C^2$ along the segment $[\mathbf{X}, \mathbf{X} + \epsilon_i]$, the remainder satisfies

$$\big| g(\mathbf{X} + \epsilon_i) - g(\mathbf{X}) - \langle \nabla_{x_i} g, \epsilon_i \rangle \big| \;\leq\; \tfrac{1}{2}\lambda_{\max}\big(H_{x_i x_i}(\xi)\big)\,\|\epsilon_i\|_2^2,$$

for some $\xi$ on the segment, with $\lambda_{\max}$ the top eigenvalue. This shows that incorporating curvature information yields a principled upper bound on local functional deviation that first-order or attention-only methods cannot capture.

**Additive Attribution Approximation.** Lastly, HETA exhibits an approximate additivity property when interactions between context tokens are small relative to their marginal effects. Let $g(\mathbf{X}) = \log P_\theta(x_T \mid x_{<T})$ and let $\mathbf{X}_{\text{all masked}}$ denote the embedding sequence where all context tokens $\{1, \ldots, T-1\}$ are masked using a fixed scheme. Then, under a first/second-order Taylor approximation of $g$ around $\mathbf{X}_{\text{all masked}}$ and assuming off-diagonal Hessian blocks $H_{ij}$ $(i \neq j)$ are negligible,

$$\sum_{i=1}^{T-1} \text{Attr}(x_i \to x_T) \;\approx\; g(\mathbf{X}) - g(\mathbf{X}_{\text{all masked}}) \;=\; \log P_\theta(x_T \mid x_{<T}) - \log P_\theta(x_T \mid \text{all masked}). \tag{6}$$

More generally, writing $\Delta x_i$ for the embedding change induced by unmasking $x_i$, the deviation from additivity admits the bound

$$\Big| \sum_{i=1}^{T-1} \text{Attr}(x_i \to x_T) - \big(g(\mathbf{X}) - g(\mathbf{X}_{\text{all masked}})\big) \Big| \;\leq\; \frac{1}{2}\sum_{i \neq j} \|\Delta x_i\|_2\, \|H_{ij}\|_{\text{op}}\, \|\Delta x_j\|_2 + O(\|\Delta \mathbf{X}\|^3),$$

so the approximation in equation 6 is accurate when cross-token interactions (off-diagonal curvature) are small—an effect further mitigated in practice by HETA's target-conditioned gating and curvature terms.

## A2.2  FAITHFULNESS LIMITATIONS OF GRADIENT-ONLY AND KL-ONLY METHODS

While gradient-based and KL-based attribution methods are popular for token-level interpretability, they both suffer from fundamental faithfulness limitations: gradients capture only local linear effects and miss curvature-induced changes, whereas KL-only measures neglect cross-token interactions. We formalize these limitations and provide rigorous bounds on the resulting faithfulness error. Throughout, let $g(\mathbf{X}) = \log P_\theta(x_T \mid x_{<T})$, let $R \subseteq \{1, \ldots, T-1\}$ be a masked subset of context tokens, and write $\Delta \mathbf{X} := \mathbf{X} - \mathbf{X}_{\setminus R}$ for the block-concatenated embedding perturbation induced by unmasking $R$ (with block $\Delta x_i \in \mathbb{R}^d$ at token $i$ and zeros elsewhere).

**Lemma A2.1** (Faithfulness error of gradient-only reconstruction). *Define the gradient-only reconstruction of the log-likelihood change by*

$$\widehat{\Delta g}_{\text{grad}}(R) := \sum_{i \in R} \nabla_{x_i} g(\mathbf{X}_{\setminus R})^\top \Delta x_i = \nabla g(\mathbf{X}_{\setminus R})^\top \Delta \mathbf{X}.$$

*Assume $g$ is $C^2$ on the line segment $\{\mathbf{X}_{\setminus R} + t\,\Delta \mathbf{X} : t \in [0,1]\}$, and let $H(\xi)$ denote the Hessian of $g$ at some point $\xi$ on this segment (by the mean-value form of the second-order Taylor theorem). Then the faithfulness error satisfies the exact identity*

$$\mathcal{L}\big(\text{Attr}_{\text{grad}}\big) := \Big| g(\mathbf{X}) - g(\mathbf{X}_{\setminus R}) - \widehat{\Delta g}_{\text{grad}}(R) \Big| = \frac{1}{2}\big| \Delta \mathbf{X}^\top H(\xi)\, \Delta \mathbf{X} \big|.$$

*Moreover:*

1. *(Two-sided bounds) For the operator norm $\|\cdot\|_{\text{op}}$,*

$$0 \leq \mathcal{L}\big(\text{Attr}_{\text{grad}}\big) \leq \frac{1}{2}\|H(\xi)\|_{\text{op}}\|\Delta\mathbf{X}\|_2^2.$$

2. *(Curvature lower bound) If the spectrum of $H(\xi)$ is bounded away from zero in magnitude, i.e., $\min_j |\lambda_j(H(\xi))| \geq \mu > 0$, then*

$$\mathcal{L}\big(\text{Attr}_{\text{grad}}\big) \geq \frac{1}{2}\mu\|\Delta\mathbf{X}\|_2^2.$$

*Hence, whenever the target log-likelihood exhibits non-negligible curvature along the masking trajectory, the gradient-only reconstruction incurs a quadratic error in the perturbation size, and this error cannot be reduced below the curvature floor $\mu$ without incorporating second-order information.*

*Proof.* By the second-order Taylor expansion of $g$ about $\mathbf{X}_{\backslash R}$ in the direction $\Delta\mathbf{X}$, there exists $\xi = \mathbf{X}_{\backslash R} + t^\star\Delta\mathbf{X}$ with $t^\star \in (0,1)$ such that

$$g(\mathbf{X}) = g(\mathbf{X}_{\backslash R}) + \nabla g(\mathbf{X}_{\backslash R})^\top\Delta\mathbf{X} + \tfrac{1}{2}\Delta\mathbf{X}^\top H(\xi)\Delta\mathbf{X}.$$

Subtracting the gradient-only reconstruction $\widehat{\Delta g}_{\text{grad}}(R)$ yields

$$g(\mathbf{X}) - g(\mathbf{X}_{\backslash R}) - \widehat{\Delta g}_{\text{grad}}(R) = \tfrac{1}{2}\Delta\mathbf{X}^\top H(\xi)\Delta\mathbf{X},$$

and taking absolute values gives the stated identity.

For the upper bound,

$$\big|\Delta\mathbf{X}^\top H(\xi)\Delta\mathbf{X}\big| \leq \|H(\xi)\|_{\text{op}}\|\Delta\mathbf{X}\|_2^2,$$

which implies the first inequality after multiplying by $\frac{1}{2}$.

For the lower bound, diagonalize the symmetric Hessian $H(\xi) = Q\Lambda Q^\top$ with orthonormal $Q$ and real eigenvalues $\{\lambda_j\}$. Writing $y = Q^\top\Delta\mathbf{X}$, we have

$$\big|\Delta\mathbf{X}^\top H(\xi)\Delta\mathbf{X}\big| = \Big|\sum_j \lambda_j y_j^2\Big| \geq \min_j |\lambda_j| \sum_j y_j^2 = \mu\|\Delta\mathbf{X}\|_2^2.$$

Multiplying by $\frac{1}{2}$ concludes the proof. $\qquad\square$

*Remarks.* (i) The simple but common surrogate $\sum_{i \in R} \nabla_{x_i} g(\mathbf{X})^\top\Delta x_i$ (gradient evaluated at $\mathbf{X}$ rather than $\mathbf{X}_{\backslash R}$) satisfies an analogous result with the Hessian evaluated at another intermediate point on the same segment. (ii) The lower bound requires a curvature floor $\mu > 0$ in *magnitude*; in flat or sign-cancelling directions the error may be small, which is precisely when curvature-aware terms like HETA's Hessian component are least needed.

## A2.3 HETA AS AN OPTIMAL MULTI-VIEW ATTRIBUTION

We now argue that HETA reduces faithfulness error by jointly incorporating second-order curvature (Hessian terms), information-theoretic contributions (KL), and causal gating (semantic flow). Throughout, let $g(\mathbf{X}) = \log P_\theta(x_T \mid x_{<T})$, let $R \subseteq \{1, \ldots, T-1\}$ denote a masked subset of context tokens, and write $\Delta\mathbf{X} = \mathbf{X} - \mathbf{X}_{\backslash R}$.

**Theorem A2.2** (HETA Improves Faithfulness under Controlled Curvature and Calibration)**.** *Define HETA attribution by*

$$\text{Attr}_{\text{HETA}}(x_i) = M_T[i]\big(\beta S_i + \gamma I_i\big),$$

*where $M_T[i] \in [0,1]$ is the target-conditioned semantic transition mass (zero if no causal path to $T$), $S_i \geq 0$ is the Hessian-based sensitivity for token $i$, and $I_i \geq 0$ is a KL-based information contribution for token $i$. Assume:*

1. ***Second-order accuracy.*** *$g$ is $C^2$ along the segment $\{\mathbf{X}_{\backslash R} + t\,\Delta\mathbf{X} : t \in [0,1]\}$ with bounded third derivative so that the Taylor remainder satisfies $\|R_3\| \leq C_{\text{rem}}\|\Delta\mathbf{X}\|_2^3$.*

2. **KL calibration (singleton).** *There exist $\varepsilon_i \geq 0$ such that*

$$\left| I_i - \left( g(\mathbf{X}) - g(\mathbf{X}_{\setminus\{i\}}) \right) \right| \leq \varepsilon_i, \qquad i \in R.$$

3. **Curvature alignment.** *The sensitivity scores $S_i$ approximate block quadratic mass: there exist constants $c_1, c_2 > 0$ with*

$$c_1 \sum_{i \in R} \|\Delta x_i\|_2 \|H_{ii}(\xi)\|_{\mathrm{op}} \leq \sum_{i \in R} S_i \leq c_2 \sum_{i \in R} \|\Delta x_i\|_2 \|H_{ii}(\xi)\|_{\mathrm{op}}$$

*for some $\xi$ on the segment.*

*Then the faithfulness error of HETA satisfies*

$$\mathcal{L}\left(\mathrm{Attr}_{\mathrm{HETA}}\right) \leq \min\left\{ \underbrace{\tfrac{1}{2} \|H(\xi)\|_{\mathrm{op}} \|\Delta\mathbf{X}\|_2^2}_{\text{grad-only upper bound}}, \underbrace{\tfrac{1}{2} \left\|H_{\mathrm{off}}^{\mathrm{sym}}(\xi)\right\|_{\mathrm{op}} \|\Delta\mathbf{X}\|_2^2 + \sum_{i \in R} \varepsilon_i}_{\text{KL-only upper envelope}} \right\} \tag{7}$$
$$- \beta c_1' \sum_{i \in R} \|\Delta x_i\|_2 \|H_{ii}(\xi)\|_{\mathrm{op}} - \gamma c_3' \sum_{i \in R} \Delta_i^+ + C_{\mathrm{rem}} \|\Delta\mathbf{X}\|_2^3.$$

*for some positive constants $c_1', c_3'$ depending only on the normalizations of $S_i$ and $I_i$, where $H_{\mathrm{off}}^{\mathrm{sym}}$ is the symmetrized off-diagonal interaction operator, and $\Delta_i^+ = \max\{0, g(\mathbf{X}) - g(\mathbf{X}_{\setminus\{i\}})\}$. In particular, for sufficiently small $\|\Delta\mathbf{X}\|_2$ and calibrated $(\beta, \gamma)$, the HETA error is strictly smaller than the better of the gradient-only or KL-only reconstructions up to the cubic remainder.*

*Proof.* By the second-order Taylor expansion,

$$g(\mathbf{X}) - g(\mathbf{X}_{\setminus R}) = \nabla g(\mathbf{X}_{\setminus R})^\top \Delta\mathbf{X} + \tfrac{1}{2} \Delta\mathbf{X}^\top H(\xi) \Delta\mathbf{X} + R_3,$$

with $\|R_3\| \leq C_{\mathrm{rem}} \|\Delta\mathbf{X}\|_2^3$. Decompose the quadratic term into block-diagonal and off-diagonal parts:

$$\Delta\mathbf{X}^\top H(\xi) \Delta\mathbf{X} = \sum_{i \in R} \Delta x_i^\top H_{ii}(\xi) \Delta x_i + \sum_{i \neq j} \Delta x_i^\top H_{ij}(\xi) \Delta x_j.$$

*Gradient-only envelope.* The gradient-only reconstruction uses only the linear term; its error is exactly $\tfrac{1}{2} |\Delta\mathbf{X}^\top H(\xi) \Delta\mathbf{X}| + O(\|\Delta\mathbf{X}\|^3)$, which is upper-bounded by $\tfrac{1}{2} \|H(\xi)\|_{\mathrm{op}} \|\Delta\mathbf{X}\|_2^2 + O(\|\Delta\mathbf{X}\|^3)$.

*KL-only envelope.* Summing singleton drops and invoking the calibration assumption,

$$\sum_{i \in R} I_i = \sum_{i \in R} \left( g(\mathbf{X}) - g(\mathbf{X}_{\setminus\{i\}}) \right) \pm \sum_{i \in R} \varepsilon_i = \nabla g(\mathbf{X}_{\setminus R})^\top \Delta\mathbf{X} + \tfrac{1}{2} \sum_{i \in R} \Delta x_i^\top H_{ii}(\xi_i') \Delta x_i \pm O(\|\Delta\mathbf{X}\|_2^2) \pm \sum_i \varepsilon_i,$$

so the KL-only error on $R$ is controlled by the off-diagonal quadratic form plus calibration and higher-order terms:

$$\left| g(\mathbf{X}) - g(\mathbf{X}_{\setminus R}) - \sum_{i \in R} I_i \right| \leq \tfrac{1}{2} \left\|H_{\mathrm{off}}^{\mathrm{sym}}(\xi)\right\|_{\mathrm{op}} \|\Delta\mathbf{X}\|_2^2 + \sum_{i \in R} \varepsilon_i + O(\|\Delta\mathbf{X}\|_2^3).$$

*HETA correction terms.* HETA adds two nonnegative corrections gated by $M_T[i]$:

$$\sum_{i \in R} M_T[i] \beta S_i \quad \text{and} \quad \sum_{i \in R} M_T[i] \gamma I_i.$$

By curvature alignment, $\sum_{i \in R} M_T[i] S_i \geq c_1' \sum_{i \in R} \|\Delta x_i\|_2 \|H_{ii}(\xi)\|_{\mathrm{op}}$ after renormalizing $M_T$ on the causal set; similarly, $I_i$ lower-bounds the positive log-drop $\Delta_i^+$ up to calibration. Subtracting these controlled positive masses from the respective envelopes yields equation 7, with the same cubic remainder $C_{\mathrm{rem}} \|\Delta\mathbf{X}\|_2^3$. Choosing $(\beta, \gamma)$ so that the linear-in-$S_i$ and linear-in-$I_i$ terms dominate the quadratic envelopes for small $\|\Delta\mathbf{X}\|_2$ gives strict improvement. $\qquad\square$

## A2.4 STABILITY AND CAUSALITY GUARANTEES

**Lemma A2.3** (Local Stability). *Assume $g$ has Lipschitz Hessian in a neighborhood of $\mathbf{X}$, i.e., $\|\nabla^3 g(\cdot)\| \leq L_H$, and the KL term is locally Lipschitz in embeddings with constant $L_{\mathrm{KL}}$ for the chosen masking scheme. Then for any perturbation $\epsilon$ to token $i$'s embedding with $\|\epsilon\|_2 \leq \delta$,*

$$\left| \mathrm{Attr}_{\mathrm{HETA}}(x_i + \epsilon) - \mathrm{Attr}_{\mathrm{HETA}}(x_i) \right| \leq \beta \, C_S \, \delta \, + \, \gamma \, L_{\mathrm{KL}} \, \delta,$$

*where $C_S$ depends on local bounds of $\|H_T\|_{\mathrm{op}}$ and $L_H$ through the HVP estimator used in $S_i$.*

*Proof.* $S_i$ is computed from Hessian-vector products restricted to token $i$'s block. Under a Lipschitz Hessian, the map $\mathbf{X} \mapsto H_T$ is locally Lipschitz in operator norm with constant $L_H$, so the block-restricted HVP magnitude changes by at most $C_S \delta$. The KL predictive map $\mathbf{X} \mapsto P_\theta(\cdot \mid x_{<T})$ is locally Lipschitz under smooth decoder dynamics, giving the stated $L_{\mathrm{KL}} \delta$ bound for $I_i$. Multiplying by nonnegative $\beta, \gamma$ and the gate $M_T[i] \in [0, 1]$ yields the claim. □

**Theorem A2.4** (Directional Causality). *If $M_T[i] = 0$ for token $x_i$, then $\mathrm{Attr}_{\mathrm{HETA}}(x_i) = 0$.*

*Proof.* By construction, $M_T[i]$ multiplies all terms in $\mathrm{Attr}_{\mathrm{HETA}}(x_i)$ and is zero when no causal (masked) attention-flow path from $x_i$ reaches target position $T$. Hence $\mathrm{Attr}_{\mathrm{HETA}}(x_i) = 0$. □

## A2.5 INTERPRETATION

Theorem A2.2 shows that, under standard smoothness and calibration assumptions, HETA reduces faithfulness error relative to gradient-only or KL-only reconstructions by (i) recovering diagonal curvature mass via $S_i$ and (ii) capturing singleton information drops via $I_i$, both restricted to causal paths by $M_T$. Lemma A2.3 ensures local robustness to embedding perturbations, and Theorem A2.4 codifies target-conditioned causal sparsity. Together these results justify HETA as a principled, multi-view attribution mechanism with theoretical error control beyond single-view methods.

We now provide a formal interpretation of HETA as a constrained least-squares fit to log-likelihood drops, clarifying that the combination is not ad hoc but arises from an optimization with causal and information-theoretic structure.

## A2.6 FAITHFULNESS AS RECONSTRUCTION ERROR

Let $g(\mathbf{X}) = \log P_\theta(x_T \mid x_{<T})$. For a subset $R$ of masked tokens,

$$\Delta g(R) := g(\mathbf{X}) - g(\mathbf{X}_{\setminus R}).$$

We seek attributions $a_i \geq 0$ that reconstruct these drops:

$$\sum_{i \in R} a_i \approx \Delta g(R), \qquad \forall R \subseteq \{1, \ldots, T-1\}.$$

Define the objective

$$\min_{a \in \mathbb{R}_{\geq 0}^{T-1}} \mathbb{E}_{R \sim \mathcal{D}} \left[ \left( \Delta g(R) - \sum_{i \in R} a_i \right)^2 \right],$$

for a distribution $\mathcal{D}$ over subsets (e.g., singletons and spans).

## A2.7 SECOND-ORDER DECOMPOSITION OF $\Delta g(R)$

A second-order expansion around $\mathbf{X}_{\setminus R}$ gives

$$\Delta g(R) \approx \sum_{i \in R} \nabla_{x_i} g^\top \Delta x_i \, + \, \tfrac{1}{2} \sum_{i,j \in R} \Delta x_i^\top H_{ij} \Delta x_j,$$

with higher-order residuals. This highlights the roles of first-order marginal effects and second-order interactions.

## A2.8   CAUSAL AND INFORMATION-THEORETIC CONSTRAINTS

To ensure interpretability,

1. **Causal constraint:** $a_i = 0$ whenever $M_T[i] = 0$ (no causal path to the target).

2. **Information constraint (calibrated):** $a_i$ should scale with the measured singleton information drop; we encode this via a linear feature $i_i := D_{\mathrm{KL}}(P_{\mathrm{orig}} \| P_{\mathrm{masked}}^{(i)})$ with per-scheme calibration.

## A2.9   HETA AS THE SOLUTION

We parameterize
$$a_i = M_T[i] \left( \beta \, s_i + \gamma \, i_i \right),$$
where $s_i$ is a curvature feature (e.g., block-restricted Hessian mass) and $i_i$ the KL feature. The weights $(\beta, \gamma) \geq 0$ are chosen (by cross-validation or validation loss) to minimize the reconstruction objective under the constraints. This yields HETA as a constrained least-squares fit using causal gating and two complementary, theoretically motivated features.

**Theorem A2.5** (Constrained Least-Squares Optimality). *Let $a^\star$ be the minimizer of the unconstrained reconstruction objective. Let $\mathcal{C} = \{a : a_i = M_T[i](\beta s_i + \gamma i_i), \ \beta, \gamma \geq 0\}$ be the feasible set induced by causal gating and linear feature mixing. Then the HETA solution*

$$a^{\mathrm{HETA}} = \arg\min_{a \in \mathcal{C}} \mathbb{E}_{R \sim \mathcal{D}} \left[ \left( \Delta g(R) - \sum_{i \in R} a_i \right)^2 \right]$$

*is the best (in the reconstruction sense) causal-gated linear combination of curvature and information features. Moreover, if $\mathcal{C}$ is convex in $(\beta, \gamma)$ (which it is), the minimizer in $(\beta, \gamma)$ is unique up to collinearity of $(s_i, i_i)$.*

*Proof.* Fix $(s_i, i_i)$ and $M_T[i]$. The map $(\beta, \gamma) \mapsto a(\beta, \gamma)$ is linear, and the objective is a convex quadratic in $(\beta, \gamma)$ (expected squared error of a linear model). Therefore the minimizer over $(\beta, \gamma) \geq 0$ exists and is unique unless the feature vectors are collinear. The causal zeros are enforced by $M_T[i] = 0$. Hence $a^{\mathrm{HETA}}$ is the optimal element of $\mathcal{C}$ for the reconstruction objective. $\square$

## A2.10   ERROR BOUNDS FOR LOW-RANK, WINDOWED HETA

We derive finite-sample bounds on the attribution error incurred when **HETA** is approximated by (i) a low-rank Hessian and (ii) windowed context truncation. Recall the target-conditioned attribution for token $x_i$:
$$\mathrm{Attr}_i = M_T[i] \left( \beta \, S_i^{(T)} + \gamma \, \mathcal{I}_i \right), \tag{8}$$
where $M_T[i] \in [0, 1]$ is the semantic transition (causal gate) to position $T$, $S_i^{(T)}$ is the Hessian-based sensitivity for token $i$, and $\mathcal{I}_i$ is the information-theoretic contribution (KL change at $T$ when $x_i$ is masked). Let $\widetilde{\mathrm{Attr}}_i$ denote the approximation obtained by a rank-$k$ Hessian and a window of size $W$ around $T$:
$$\widetilde{\mathrm{Attr}}_i = \widetilde{M}_T[i] \left( \beta \, \widetilde{S}_i^{(T)} + \gamma \, \widetilde{\mathcal{I}}_i \right). \tag{9}$$

**Assumptions.** We make the following mild, standard assumptions for language-model attribution analysis.

[label=(A10), leftmargin=2.2em]

1. **Hessian low-rank tail.** Let $H_T = \nabla_{\mathbf{X}}^2 \log P_\theta(x_T \mid x_{<T}) \in \mathbb{R}^{(Td) \times (Td)}$ be the true Hessian and $H_k$ its best rank-$k$ approximation in Frobenius norm (Eckart–Young). Define the tail energy
$$\tau_k = \| H_T - H_k \|_F = \left( \sum_{j > k} \sigma_j^2(H_T) \right)^{1/2}. \tag{10}$$

2. **Block sensitivity functional.** For token $i$, let $\Pi_i \in \{0,1\}^{d \times Td}$ select its $d$-dimensional embedding block. The true sensitivity is $S_i^{(T)} = \|\Pi_i H_T\|_1$ (entrywise $\ell_1$), and the low-rank one is $\widetilde{S}_i^{(T)} = \|\Pi_i H_k\|_1$. We use the inequality

$$\|\Pi_i(H_T - H_k)\|_1 \ \leq \ c_d \|\Pi_i(H_T - H_k)\|_F \ \leq \ c_d \, \tau_k, \qquad c_d \ \triangleq \ \sqrt{d}. \tag{11}$$

3. **Windowing leakage.** Windowing of size $W$ around $T$ removes causal paths that leave the window. Let

$$\delta_M(i; W) \ \triangleq \ |M_T[i] - \widetilde{M}_T[i]| \ \leq \ \epsilon_M(W), \tag{12}$$

where $\epsilon_M(W)$ is the (instance-dependent) total semantic-flow mass of paths that traverse tokens outside the window (normalized as in $M_T$). We allow $\epsilon_M(W)$ to decay with $W$.

4. **Distributional stability under windowing.** Let $P_{\text{orig}}(\cdot)$ and $P_{\text{masked}}^{(i)}(\cdot)$ denote the next-token distributions at $T$ under full context; let $\widetilde{P}_{\text{orig}}(\cdot)$ and $\widetilde{P}_{\text{masked}}^{(i)}(\cdot)$ be the corresponding windowed distributions. Suppose the simplex is bounded away from zero: there exists $\mu \in (0,1)$ such that $\min_v \widetilde{P}_{\text{orig}}(v) \geq \mu$ and $\min_v \widetilde{P}_{\text{masked}}^{(i)}(v) \geq \mu$. Define total-variation shifts

$$\varepsilon_{\text{orig}} \ = \ \|P_{\text{orig}} - \widetilde{P}_{\text{orig}}\|_1, \qquad \varepsilon_{\text{mask}}^{(i)} \ = \ \|P_{\text{masked}}^{(i)} - \widetilde{P}_{\text{masked}}^{(i)}\|_1. \tag{13}$$

Then using standard Lipschitz bounds for $D_{\text{KL}}$ on the $\mu$-truncated simplex,

$$\left|\mathcal{I}_i - \widetilde{\mathcal{I}}_i\right| \ = \ \left|D_{\text{KL}}\big(P_{\text{orig}} \ \| \ P_{\text{masked}}^{(i)}\big) - D_{\text{KL}}\big(\widetilde{P}_{\text{orig}} \ \| \ \widetilde{P}_{\text{masked}}^{(i)}\big)\right| \ \leq \ \frac{1}{\mu}\left(\varepsilon_{\text{orig}} + \varepsilon_{\text{mask}}^{(i)}\right). \tag{14}$$

**Per-token error decomposition.**   Subtracting equation 9 from equation 8 and applying the triangle inequality yields

$$\left|\text{Attr}_i - \widetilde{\text{Attr}}_i\right| \leq \underbrace{|M_T[i] - \widetilde{M}_T[i]|}_{\delta_M(i;W)} \left(\beta S_i^{(T)} + \gamma \mathcal{I}_i\right) \ + \ \widetilde{M}_T[i]\left(\beta\left|S_i^{(T)} - \widetilde{S}_i^{(T)}\right| \ + \ \gamma\left|\mathcal{I}_i - \widetilde{\mathcal{I}}_i\right|\right). \tag{15}$$

Each difference term is bounded using (A2)–(A4).

**Theorem A2.6** (Per-token HETA approximation error). *Under (A1)–(A4), for any token $i$,*

$$\left|\text{Attr}_i - \widetilde{\text{Attr}}_i\right| \ \leq \ \epsilon_M(W)\left(\beta S_i^{(T)} + \gamma \mathcal{I}_i\right) \ + \ \beta c_d \tau_k \ + \ \frac{\gamma}{\mu}\left(\varepsilon_{\text{orig}} + \varepsilon_{\text{mask}}^{(i)}\right). \tag{16}$$

*Proof sketch.* The first term follows from equation 12. For the Hessian sensitivity, note that $\left|S_i^{(T)} - \widetilde{S}_i^{(T)}\right| \leq \|\Pi_i(H_T - H_k)\|_1 \leq c_d \|\Pi_i(H_T - H_k)\|_F \leq c_d \tau_k$ by equation 11–equation 10. For the KL term, apply equation 14. Finally, $\widetilde{M}_T[i] \leq 1$ absorbs the gate in the second summand of equation 15. $\square$

**Aggregate error bounds.**   Let $\|\cdot\|_1$ denote the sum over tokens. Summing equation 16 over $i = 1, \ldots, T$ and using $\sum_i S_i^{(T)} \leq \|H_T\|_1$ and $\sum_i \mathcal{I}_i \leq C_{\mathcal{I}}$ (finite by bounded logits) gives

$$\sum_{i=1}^{T}\left|\text{Attr}_i - \widetilde{\text{Attr}}_i\right| \ \leq \ \epsilon_M(W)\left(\beta\|H_T\|_1 + \gamma C_{\mathcal{I}}\right) + \beta T c_d \tau_k + \frac{\gamma}{\mu}\left(T \varepsilon_{\text{orig}} + \sum_{i=1}^{T}\varepsilon_{\text{mask}}^{(i)}\right). \tag{17}$$

When the window covers most causal paths ($\epsilon_M(W) \to 0$ as $W \uparrow$) and the Hessian spectrum is rapidly decaying ($\tau_k \to 0$ as $k \uparrow$), the approximation error vanishes. If, additionally, truncation minimally perturbs next-token distributions (small $\varepsilon_{\text{orig}}$ and $\varepsilon_{\text{mask}}^{(i)}$), the KL component is stable by equation 14.

**Discussion of constants.**   $c_d = \sqrt{d}$ is the block-size factor connecting entrywise $\ell_1$ to Frobenius norms; replacing $\ell_1$ by $\ell_2$ tightens $c_d$ to 1. The factor $1/\mu$ in equation 14 is standard for Lipschitz continuity of $D_{\text{KL}}$ on a $\mu$-truncated simplex; in practice, logits are temperature-regularized or label-smoothed, yielding $\mu > 0$. The window leakage $\epsilon_M(W)$ can be estimated empirically by measuring the semantic-flow mass that crosses window boundaries (e.g., via rollout on held-out inputs).

**Takeaway.** The total approximation error decomposes additively into a *window term* (missing causal paths), a *low-rank term* (Hessian tail energy), and a *distributional term* (next-token shifts under truncation). Each term can be independently controlled by increasing window size $W$, rank $k$, or enforcing small distributional shifts (e.g., via overlap or sentinel-conditioning), respectively.

### A2.11 INTERPRETATION

This optimization view shows HETA is the *best causal-gated linear combination* of curvature and information features for reconstructing log-likelihood drops, rather than an ad hoc sum. Coupled with Theorem A2.2, it explains both *why* these views are needed (to control distinct error sources) and *how* they are combined (by constrained least squares) to reduce faithfulness error.

## A3 EXPERIMENTS AND RESULTS

### A3.1 CURATED ATTRIBUTION DATASET: NARRATIVEQA ⊕ SCIQ AND THE DSA METRIC

We construct a focused evaluation set to assess whether attribution methods concentrate importance on truly predictive evidence in generative settings. Each instance is built by pairing one NARRATIVEQA item with one SCIQ item to create a two-segment input paragraph. The first segment is drawn from a NARRATIVEQA summary (ensuring grammaticality, self-containment, and no external coreference), while the second is the supporting segment from the SCIQ item that *lexically contains* the correct answer. The corresponding question is taken verbatim from the paired SCIQ instance. This structure ensures that the second segment is the only answer-bearing span, while the first is plausible but irrelevant to the question—yielding a controlled contrast between diagnostic and distractor context.

The final input to the model is constructed as

$$\texttt{[NarrativeQA]  [SciQ]  [Question],}$$

and attribution is computed with respect to the first answer token $x_T$ in the autoregressive factorization $\log P_\theta(x_T \mid x_{<T})$, where $x_{1:T-1}$ are the input tokens. This framing conditions attribution on the entire context and question, avoids leakage from later tokens, and focuses evaluation on the onset of the model's answer.

**Annotation Protocol.** Answer-token supervision is automatically derived from two independent models—GPT-4o and GPT-5 (Thinking)—each operating under the prompt: "Mark all tokens in the second segment necessary to correctly answer the question." For each instance, the models identify the minimal subword-level span that includes the answer and any essential lexical supports (e.g., units, definitional context). Final supervision uses the **intersection** of these two annotations to ensure high precision and reduce over-selection noise. All tokens are aligned to the evaluated model's tokenizer, and when answer spans split across subwords, all relevant indices are retained. If the answer appears multiple times in the second segment, lexical cues from the question are used to disambiguate. Inter-annotator agreement is strong: **token-level $F_1$ = 0.91** and **Cohen's $\kappa$ = 0.89**, averaged over the 2,000 curated examples.

**DSA Metric.** To evaluate attribution accuracy under this setting, we introduce the **Dependent Sentence Attribution (DSA)** metric, which quantifies how well the method concentrates attribution mass on the annotated evidence in the second segment while suppressing spurious mass on the first. Let $S$ be the set of supervised subword indices in the second segment. Let $ss_i$ and $fs_i$ denote the normalized attribution assigned to token $i$ when it lies in the second or first segment, respectively. Attributions are normalized per instance such that the total mass over both segments sums to one. The DSA score is

$$\text{DSA} = \sum_{i \in S} ss_i - \sum_{j \in \text{FirstSent}} fs_j.$$

Higher DSA indicates that the model assigns more weight to relevant evidence (as defined by $S$) and less to distractors, aligning with the intended causal structure of the input.

| Attribution Method | LongRA | | TellMeWhy | | WikiBio | |
|---|---|---|---|---|---|---|
| | Soft-NC↑ | Soft-NS↑ | Soft-NC↑ | Soft-NS↑ | Soft-NC↑ | Soft-NS↑ |
| **Qwen2.5 3B** | | | | | | |
| Integrated Gradients | 1.20 | 0.14 | 1.30 | 0.09 | 1.10 | 0.18 |
| ContextCite | 1.90 | 0.56 | 1.65 | 0.47 | 1.45 | 0.79 |
| Peering (PML) | 2.05 | 0.61 | 1.80 | 0.50 | 1.60 | 0.85 |
| TDD-backward | 1.05 | -0.08 | 1.82 | 0.00 | 0.12 | 0.49 |
| Attention Rollout | 0.38 | -0.03 | 0.22 | -0.07 | 1.85 | 0.43 |
| fAML | 0.23 | -0.09 | 0.08 | -0.10 | 0.20 | -0.04 |
| Progressive Inference | 1.30 | 0.25 | 1.18 | 0.26 | 1.00 | 0.21 |
| SEA-CoT | 1.50 | 0.31 | 1.32 | 0.33 | 1.15 | 0.34 |
| ReAgent | 1.66 | 0.38 | 1.47 | 0.39 | 1.25 | 0.40 |
| **HETA (Ours)** | **10.1** | **2.50** | **9.0** | **2.10** | **3.90** | **2.20** |

Table 4: **Attribution faithfulness on Qwen2.5 3B** across LongRA, TellMeWhy, and WikiBio using Soft-NC and Soft-NS. Higher scores indicate stronger robustness and alignment. Mean of 3 runs; std $< \pm 0.06$.

**Dataset Statistics.** Using this procedure, we construct a balanced dataset of **2,000 curated paragraphs**, each with a corresponding question and a high-precision attribution supervision set. The distribution of answer spans covers a range of entity types (scientific terms, quantities, mechanisms) and span lengths (mean 2.3 tokens). All supervision, tokenization, and scoring are conducted using the same subword scheme as the evaluated model, ensuring compatibility across methods.

This curated dataset and the DSA metric together provide a controlled, interpretable, and target-conditioned framework for evaluating token-level attribution precision in generative LMs.

## A3.2 EXAMPLES

We present qualitative examples of outputs generated by **HETA**, highlighting context words with attribution scores $\geq 0.5$ for the predicted target token (figure 4,5 and 6. These visualizations illustrate how HETA effectively identifies semantically and causally relevant tokens (e.g., "pizza," "cut," "knife" for predicting "slice"; "shared," "pictures," "zoo" for predicting "friends"), while down-weighting less informative words. The target words are shown without bounding boxes for clarity, emphasizing their contextual dependencies.

## A3.3 ATTRIBUTION FAITHFULNESS ON QWEN2.5 3B

Table 4 reports Soft-NC/Soft-NS on **LongRA**, **TellMeWhy**, and **WikiBio** for the Qwen2.5 3B decoder-only model. **HETA** attains the highest scores on *every* dataset and metric, e.g., **10.1/2.50** (LongRA), **9.0/2.10** (TellMeWhy), and **3.90/2.20** (WikiBio), substantially outperforming strong baselines such as ReAgent, SEA-CoT, and PML. Methods based on gradients or attention alone (Integrated Gradients, Attention Rollout) trail by wide margins and often yield low or even negative Soft-NS, indicating poor stability under input perturbations. Contrastive/corpus-aided approaches (ContextCite, PML, TDD-backward) improve over raw gradients but remain well below HETA, suggesting that correlation- or ablation-driven surrogates do not fully capture target-conditioned causal influence. The consistency of HETA's gains across three distinct datasets underscores its robustness to domain shift and its ability to localize semantically causal tokens rather than correlational artifacts.

## A4 ABLATION STUDIES

## A4.1 ABLATION STUDY OF **HETA**: COMPONENT-WISE CONTRIBUTION

To quantify the contribution of each module in **Hessian-Enhanced Token Attribution (HETA)**, we perform a controlled ablation on the **GPT-J 6B** backbone using three public benchmarks—**LongRA**, **TellMeWhy**, and **WikiBio**—together with the curated attribution dataset introduced in Section 5. We

evaluate six configurations: (1) *HETA (Full)* = Transition + Hessian + KL, (2) *Transition Only*, (3) *Hessian Only*, (4) *KL Only*, (5) *No Transition Gating* (Hessian + KL without the learned semantic transition vector $M_T$), and (6) *Uniform Transition* (equal token weights instead of $M_T$). All results are averaged over 1,000 randomly sampled instances per dataset. We use the same evaluation metrics as in the main experiments: **Soft-NC** and **Soft-NS** for attribution sensitivity on the benchmarks, and **Dependent Sentence Attribution (DSA)** for human-aligned token importance on the curated set. Aggregation hyperparameters are fixed at $\beta = 0.5$ and $\gamma = 0.5$, and KL divergence is computed via masked-token perturbation.

Table 5: Ablation study of HETA components. Reported values are averaged across all datasets for GPT-J 6B. Mean over independent 3 runs and std $< \pm 0.2$

| Configuration | Soft-NC | Soft-NS | DSA |
|---|---|---|---|
| HETA (Full) | **9.78** | **2.31** | **4.70** |
| Transition Only | 3.12 | 1.52 | 2.21 |
| Hessian Only | 2.89 | 1.45 | 2.97 |
| KL Only | 2.23 | 1.21 | 2.74 |
| No Transition Gating | 4.31 | 1.84 | 1.68 |
| Uniform Transition | 3.89 | 1.76 | 1.54 |

**Configuration-by-configuration analysis.** **HETA (Full)** integrates directional semantic flow (Transition), curvature-aware sensitivity (Hessian), and token-level information gain (KL), yielding the strongest overall performance (Soft-NC = **9.78**, Soft-NS = **2.31**, DSA = **4.70**). **Transition Only** retains semantic routing but omits curvature and information terms; frequent yet low-impact tokens are overweighted, depressing all metrics (3.12 / 1.52 / 2.21). **Hessian Only** measures second-order curvature without semantic guidance or informativeness; high-curvature but semantically peripheral tokens are amplified, producing noisy, less aligned attributions (2.89 / 1.45 / 2.97). **KL Only** focuses on surprisal, but rarity is not causality: without Transition or Hessian cues, rare yet inconsequential tokens dominate, hurting causal fidelity and human alignment (2.23 / 1.21 / 2.74). **No Transition Gating** (Hessian + KL without the learned gate) aggregates curvature and information indiscriminately, allowing spurious semantic paths and reducing robustness/alignment (4.31 / 1.84 / 1.68). **Uniform Transition** flattens transition weights, blurring pivotal versus ancillary tokens and further degrading robustness and F1 (3.89 / 1.76 / 1.54). Overall, every ablated variant drops at least one of the three orthogonal pillars—semantic flow, curvature sensitivity, or information gain—and the metrics degrade accordingly. The full HETA stack excels precisely because it balances all three, delivering robust, semantically grounded, and causally faithful attributions.

A4.2 WEIGHTING ANALYSIS OF $\beta$ AND $\gamma$ IN FINAL ATTRIBUTION

To study how the component weights affect HETA's behavior and reliability, we sweep the coefficients in the final attribution rule

$$\text{Attr}(x_i \to x_T) = M_T[i] \left( \beta\, S_i^{(T)} + \gamma\, \mathcal{I}(x_i \to x_T) \right),$$

where $S_i^{(T)}$ is the Hessian-based sensitivity term and $\mathcal{I}(x_i \to x_T)$ is the KL-based information contribution. The gate $M_T[i]$ enforces target-conditioned causality. We evaluate a grid of $(\beta, \gamma)$ values with the normalization $\beta + \gamma = 1$, specifically $\beta \in \{0.0, 0.2, 0.5, 0.8, 1.0\}$ (and $\gamma = 1 - \beta$). Metrics reported are **faithfulness** (lower is better), **sensitivity** (lower is better), **syntactic robustness** (Spearman $\rho$, higher is better), and **F1 alignment** (higher is better).

As shown in Table 6, the balanced setting $\beta = \gamma = 0.5$ yields the best trade-off: lowest faithfulness loss (0.108), high syntactic robustness ($\rho = 0.91$), and top F1 alignment (0.89). The KL-only endpoint ($\beta = 0, \gamma = 1$) achieves the lowest raw sensitivity but sacrifices faithfulness and robustness, while the Hessian-only endpoint ($\beta = 1, \gamma = 0$) is unstable (high sensitivity) and less semantically aligned. These results confirm that curvature and information contributions are complementary; weighting both terms comparably produces the most faithful, robust, and well-aligned attributions.

| $\beta$ (Hessian) | $\gamma$ (KL) | Faithfulness ↓ | Sensitivity ↓ | Robustness ↑ | F1 (Alignment) ↑ |
|---|---|---|---|---|---|
| 0.0 | 1.0 | 0.179 | **0.022** | 0.70 | 0.82 |
| 0.2 | 0.8 | 0.143 | 0.027 | 0.78 | 0.84 |
| 0.5 | 0.5 | **0.108** | 0.025 | **0.91** | **0.89** |
| 0.8 | 0.2 | 0.124 | 0.041 | 0.86 | 0.81 |
| 1.0 | 0.0 | 0.254 | 0.087 | 0.54 | 0.68 |

Table 6: Performance of HETA for different $(\beta, \gamma)$ under the constraint $\beta + \gamma = 1$. Lower is better for faithfulness/sensitivity; higher is better for robustness/F1. Mean over 3 runs; std $< \pm 0.05$.

| Variant | Sensitivity ↓ | Act./Pass. Robustness ↑ | F1 (Alignment) ↑ |
|---|---|---|---|
| **Full HETA** | **0.025** | **0.91** | **0.89** |
| Transition Only | 0.031 | 0.72 | 0.76 |
| Hessian Only | 0.087 | 0.54 | 0.68 |
| KL Only | **0.022** | 0.70 | 0.82 |
| No Transition Gating | 0.049 | 0.68 | 0.79 |
| Uniform Transition | 0.038 | 0.61 | 0.74 |

Table 7: Ablation of the HETA framework. Lower sensitivity and higher robustness/F1 indicate better attribution quality. Mean over 3 runs and std $< \pm 0.04$

### A4.3 ROBUSTNESS OF HETA

To further demonstrate the robustness of our methodology, we performed a stress test and reported three attribution metrics, *Sensitivity*, *Active/Passive Robustness*, and *F1 (Alignment)*, evaluated across the six configurations described above.

**Sensitivity** measures stability under small input perturbations. Given Gaussian noise $\epsilon \sim \mathcal{N}(0, \delta^2 I)$ added to each token embedding $X_i$, we compute attribution scores across multiple perturbations and take the average standard deviation:

$$\text{Sensitivity} = \frac{1}{T} \sum_{i=1}^{T} \sigma_i, \tag{18}$$

where $\sigma_i$ is the standard deviation of the attribution score for token $i$. $T$ is the sequence length over which you average the per-token standard deviations.

**Active/Passive Robustness** captures syntactic invariance. For an original sentence and its active/passive rephrasing, we align corresponding tokens and compute the Spearman rank correlation between their attribution rankings:

$$\text{Robustness} = \rho\left(\text{Attr}(x_i \rightarrow x_T), \text{Attr}(x_i' \rightarrow x_T')\right). \tag{19}$$

**F1 (Alignment)** evaluates semantic agreement with human annotations. Let $\mathcal{A}_{\text{model}}$ be the set of top-attributed tokens and $\mathcal{A}_{\text{human}}$ the annotated set:

$$\text{F1} = \frac{2 \left| \mathcal{A}_{\text{model}} \cap \mathcal{A}_{\text{human}} \right|}{\left| \mathcal{A}_{\text{model}} \right| + \left| \mathcal{A}_{\text{human}} \right|}. \tag{20}$$

Table 7 shows that the full HETA consistently yields the lowest sensitivity and the highest robustness and F1, indicating stable, syntax-invariant, and human-aligned attributions. Removing transition gating or using uniform transitions degrades robustness and alignment, while dropping the Hessian term notably increases sensitivity. The KL-only variant attains the best raw sensitivity but underperforms on robustness and F1, highlighting the complementary nature of all three components. Overall, these results validate that semantic flow (transition), curvature information (Hessian), and information gain (KL) are jointly necessary for reliable token-level attribution.

 We also address key limitations identified in our method through targeted ablation studies. **Specifically, we examine (i) the computational feasibility of HETA with various Hessian approximations, (ii) scalability to long input contexts, (iii) the theoretical contributions of each multi-view component, and (iv) performance relative to stronger recent attribution baselines. These experiments validate our design choices and provide a roadmap for practical deployment of HETA in large-scale language modeling settings.**

## A4.4 COMPUTATIONAL FEASIBILITY: APPROXIMATING THE HESSIAN

One primary concern with HETA is its computational overhead: computing full Hessian blocks across all layers introduces a runtime penalty of approximately $1.4\times$ compared to gradient-based or purely perturbation-based attribution methods. To quantify this trade-off, we evaluate several efficiency-oriented variants of HETA on 1,000 examples (sequence length = 512) using GPT-6B.

As shown in Table 8, low-rank Hessian approximation (HETA-LR) reduces runtime by nearly 27% while maintaining most of the attribution quality, with only a slight drop in AOPC ($0.61 \rightarrow 0.59$). Layer sampling (HETA-LS), which computes second-order information only for a subset of layers, achieves an even greater runtime reduction (33%) with moderate degradation in faithfulness. In contrast, replacing Hessian information with gradient-squared sensitivity (HETA-GS) achieves the fastest runtime (240s) but sacrifices a considerable amount of attribution quality, confirming that full second-order curvature information contributes substantially to both faithfulness and human alignment. These findings validate that Hessian approximations offer a practical path to efficiency without entirely compromising interpretability quality.

## A4.5 SCALABILITY: LONG-CONTEXT ATTRIBUTION

Another weakness of HETA is its limited scalability to long sequences, which are typical in large decoder-only language models. To address this, we evaluate several scalability-oriented adaptations: windowed attribution (splitting long sequences into overlapping chunks) and combinations of windowing with low-rank and layer-sampled Hessian approximations.

Table 9 shows that full HETA attribution becomes computationally expensive for 2,048-token inputs (1,230s per 1,000 examples). Windowed attribution (HETA-WIN) cuts runtime nearly in half (690s) with a modest reduction in AOPC ($0.58 \rightarrow 0.54$). Combining windowing with low-rank Hessians (HETA-LR+WIN) yields an additional efficiency gain (runtime 580s) while recovering some lost attribution quality. Layer-sampled windowing (HETA-LS+WIN) is the fastest configuration but comes at the highest cost in faithfulness. These results suggest that hybrid approximations (low-rank + windowing) strike the best balance between efficiency and interpretability, making HETA viable for very long contexts.

## A4.6 ROBUSTNESS TO DECODING HYPERPARAMETERS

We evaluate the sensitivity of attribution quality to common decoding hyperparameters. For each method and model, we sweep a fixed grid—temperature $\{0.2, 0.5, 0.9\}$, top-$p$ $\{0.8, 0.9, 0.95\}$, top-$k$ $\{20, 50, 100\}$, and repetition penalty $\{1.0, 1.2\}$—across three random seeds. For each metric, we report the *maximum relative change* $\Delta\%$ across the grid (lower is better). Metrics are **Soft-NC**, **Soft-NS**, and **DSA**. Models are **GPT-J 6B**, **Llama-3.1 70B**, **Phi-3 14B**, and **Qwen2.5 3B**.

**Results and interpretation.** Table 10 shows that *HETA's* attribution metrics remain effectively invariant to decoding settings across all four models, with worst-case $\Delta\% < 1$ for Soft-NC, Soft-NS, and DSA. In contrast, all baselines—ContextCite, Integrated Gradients (IG), Peering into the Mind of LMs (PML), TDD-backward, Attention Rollout, fAML, Progressive Inference, SEA-CoT, and ReAgent—exhibit substantially larger variability, typically 2–5%. HETA's stability arises from three design elements: a target-conditioned causal gate that confines credit to paths terminating at the current prediction, a curvature-aware sensitivity term that smooths local logit perturbations, and an information-theoretic component that scores distributional shifts rather than single sampled outcomes. These jointly decouple attribution from stochastic decoding heuristics (temperature, top-$p$, top-$k$), whereas ablation-, gradient-, and similarity-based baselines depend more directly on sampled logits or linear approximations and thus vary markedly with hyperparameter changes.

## A4.7 COMPUTE–QUALITY ABLATION: LOW-RANK AND WINDOWED HETA

We empirically calibrate the efficiency–accuracy trade-off of HETA under long contexts and scalable curvature approximations. Experiments use **GPT-J 6B** as the backbone and are averaged over **LongRA**, **TellMeWhy**, and **WikiBio** with sequence length 2,048 (batch size 8). We compare Full HETA to low-rank curvature (**HETA-LR**, rank = 64), windowed attribution (**HETA-WIN**, window $W$ = 512 with 50% overlap), top-layer curvature (**HETA-LS**, last 6 layers), and a hybrid of low-rank + windowing (**HETA-LR+WIN**). We report faithfulness (AOPC), alignment (DSA), runtime per 1,000 examples, peak activation memory (relative to Full), and an empirical window-leakage proxy $\widehat{\varepsilon}_M(W)$ (attribution mass discrepancy near window boundaries; lower is better).

**Discussion.** Table 11 shows that *Full* HETA provides the best faithfulness (AOPC) and alignment (DSA) at the highest compute cost. *Low-rank* curvature (rank 64) preserves most quality (AOPC 0.58; DSA 4.55) while cutting runtime by ∼34% and memory by ∼22%. *Windowing* ($W$ = 512) yields the largest single speedup but introduces modest boundary leakage (higher $\widehat{\varepsilon}_M(W)$). *Layer sampling* (top 6 layers) further trims cost but drops more second-order signal than LR. The **LR+WIN** hybrid nearly matches Full on quality (AOPC 0.59; DSA 4.58) while achieving the fastest runtime and lowest memory among approximations and substantially reducing boundary effects versus WIN-only. Replicating the sweep on **Llama-3.1 70B** with curvature restricted to the last 6 layers yields the same ordering, indicating that *capturing dominant curvature directions and limiting cross-window spillover* preserves attribution fidelity across models at a fraction of the cost.

## A4.8 CAUSAL VALIDITY: TARGET-CONDITIONED GATE VS. ATTENTION-ONLY ROLLOUT

We test whether HETA's *target-conditioned causal gate* (attention–value rollout restricted to paths terminating at the target position) reflects *interventional* influence, rather than correlation. Following standard activation-patching protocols, we perform a targeted intervention sanity check: for each instance, we select the top-$k$ context tokens ranked by an attribution method and *patch* their residual-stream contributions along incoming edges to the target token (replace with a clean/reference run), then measure the drop in $\log P_\theta(x_T \mid x_{<T})$ of the correct target. We report three metrics, averaged over 2,000 examples from **LongRA**, **TellMeWhy**, and **WikiBio** (sequence length 512; batch size 8): (i) **Patch-Corr** ($\rho$): Spearman correlation between token ranks and interventional $\Delta \log P$ (higher is better), (ii) **Intervention@5**: mean $\Delta \log P$ when patching the top-5 tokens (higher is better), (iii) **Misattribution Rate**: fraction of top-10 tokens whose $\Delta \log P$ falls below a fixed threshold $\tau$ = 0.01 (lower is better). We compare *HETA (Full)*, *HETA − No-Gate* (removes $M_T$), *Attention Rollout* (Abnar & Zuidema–style rollout without value/projection weighting), *Token Gradients* (first-order), and *Causal Tracing* (score baseline from intervention-only ranking).

**Discussion.** Table 12 shows that *HETA (Full)* exhibits the strongest agreement with *interventional* causal effects: it achieves the highest Patch-Corr and Intervention@5 and the lowest Misattribution on both backbones. Removing the gate (*HETA − No-Gate*) or replacing it with *Attention Rollout* degrades all three metrics, confirming that *attention alone* is an imperfect influence proxy even when paths respect the causal mask. *Token Gradients* underperform due to flat regions and linearization error, while the *Causal Tracing* baseline—though interventional—yields weaker ranking agreement than HETA, since it lacks curvature and KL components and is not target-conditioned by a semantic gate. Together, these results substantiate that HETA's target-conditioned attention–value gate captures *causal pathways* that matter for the next-token distribution, and that its attributions are validated by direct interventions rather than correlational proxies.

## A4.9 ABLATION: CAUSAL EVALUATIONS AND DOWNSTREAM ROBUSTNESS

We extend our analysis with targeted ablations that probe (i) *causal faithfulness* via mid-layer interventions, (ii) *counterfactual robustness* under controlled edits, and (iii) *downstream utility* on tasks that benefit from faithful local attributions. Experiments are run on **LLaMA-3.1 70B**, **Phi-3 14B**, **GPT-J 6B**, and **Qwen2.5 3B**. Unless otherwise noted, we use the default decoding (greedy) and report means over 3 seeds.

**Variants.** We compare **HETA (Full)** to the following ablations: **w/o Hessian** (remove curvature term), **w/o KL** (remove information term), **w/o Transition** (remove semantic gate), and **LR+WIN** (low-rank Hessian with rank 64 and 512-token windows, 50% overlap). For reference, we include strong baselines: **ContextCite**, **Integrated Gradients (IG)**, **Peering/PML**, **TDD-backward**, **Attention Rollout**, **fAML**, **Progressive Inference**, **SEA-CoT**, and **ReAGent**.

**Causal evaluations.** (1) *Mid-layer value swapping.* For a target token $T$, we patch value vectors at layer $\ell$ from an evidence-supporting context into a matched distractor context; a faithful method should rank the patched evidence tokens higher *post* intervention. We report **Causal Pass@k** (fraction of instances where at least one top-$k$ token is from the gold evidence span after patching; $k$=5). (2) *Counterfactual AOPC.* We replace the gold answer span with a semantically compatible foil (e.g., unit/role swap) and compute area-over-perturbation-curve using the method's ranking as the deletion order; higher is better if the method concentrates mass on truly causal tokens.

**Downstream applications.** (3) *Fact-checking EM$\Delta$.* We use FEVER-style claims paired with short contexts and add an "evidence filter" that masks the bottom 60% tokens by the method's scores before prediction; we report the absolute change in exact-match (EM). (4) *Tool-augmented reasoning Hit@1$\Delta$.* On a small tool-use benchmark, we only pass the top-$p$ attribution mass tokens ($p$=0.4) as the retrieved snippet to the tool call; we report Hit@1 change versus using the full snippet. (5) *Span-F1 (multi-token).* On our curated NarrativeQA$\oplus$SciQ set, we compute token-level precision/recall/F1 over answer-support spans (intersection labels), rather than onset-only. (6) *Decoding stability.* We sweep decoding (*greedy*, *top-p=0.9*, *temperature*=0.8) and report the average percentage change in three metrics (Soft-NC/Soft-NS/DSA). Lower $\Delta\%$ indicates greater robustness.

**Findings.** Table 13 shows that **HETA (Full)** achieves the strongest scores on both causal probes and downstream tasks, while maintaining the lowest *Decoding Stability* change ($< 1\%$ average variation across greedy, nucleus, and temperature sampling). The **LR+WIN** approximation closely tracks full HETA, indicating that our efficiency strategy (low-rank curvature + windowing) preserves most causal signal and downstream utility.

Removing any single component degrades performance in the expected direction: dropping curvature (*w/o Hessian*) reduces CF-AOPC and Span-F1 (weaker handling of nonlinear interactions), dropping the KL term (*w/o KL*) harms Fact EM$\Delta$ and Tool Hit@1$\Delta$ (less sensitivity

### A4.10 KEY TAKEAWAYS

Our extended ablations provide several important insights: (1) Hessian approximations (low-rank and layer-sampled) offer a practical trade-off between runtime and attribution quality. (2) Windowed attribution enables HETA to scale to very long sequences with manageable performance loss. (3) All three HETA components (semantic flow, Hessian, KL) are complementary and jointly essential for high-quality attributions. (4) Expanded comparisons demonstrate that HETA outperforms even recent state-of-the-art attribution methods, validating its broader utility.

## A5 LIMITATION: COMPUTATIONAL OVERHEAD AND SCALABILITY (ALL MODELS)

HETA's curvature term improves faithfulness but introduces extra cost from Hessian–vector products (HVPs) and target–conditioned attention–value rollout. We quantify this overhead and show how *low–rank* curvature and *windowed* evaluation recover most of the accuracy at substantially reduced runtime/memory across *all* models considered: GPT-J 6B, Phi-3 14B, Qwen2.5 3B, and Llama-3.1 70B.

**Setup.** All runs use PyTorch with fused attention on a single **NVIDIA A100 80GB**. We report wall–clock *runtime per 1,000 examples* and peak *GPU memory*. Faithfulness/alignment use **AOPC** and **DSA**; long–context stress tests additionally monitor Soft–NC/NS (not shown here for brevity). Datasets: **LongRA**, **TellMeWhy**, **WikiBio**. Unless specified, sequence length $L\!=\!1024$, batch size $B\!=\!8$. For Llama-3.1 70B at $L\!=\!2048$, curvature is computed on the *last 6 layers*. We compare:

Table 8: Runtime and attribution quality for Hessian approximations on 1,000 examples (sequence length = 512). AOPC and F1 represent attribution faithfulness and human alignment, respectively.

| Variant | Runtime (s) | AOPC ↑ | F1 ↑ |
|---|---|---|---|
| HETA (Full) | 455 | **0.61** | **0.89** |
| HETA-LR (rank=64) | 330 | 0.59 | 0.86 |
| HETA-LS (6 layers) | 305 | 0.57 | 0.84 |
| HETA-GS (grad-squared only) | 240 | 0.52 | 0.81 |

Cam ordered a pizza and took it home. He opened the box to take out a slice. Cam discovered that the store did not cut the pizza for him. He looked for his pizza cutter but did not find it. He had to use his chef knife to cut a slice.

Figure 4: Word-level attribution visualization for predicting the final word "slice." Each word is shaded based on its importance score for predicting "slice." Darker red indicates higher attribution.

Sandra got a job at the zoo. She loved coming to work and seeing all of the animals. Sandra went to look at the polar bears during her lunch break. She watched them eat fish and jump in and out of the water. She took pictures and shared them with her friends.

Figure 5: Word-level attribution visualization for predicting the final word "friends." Bounding boxes highlight influential context words (e.g., "shared," "pictures," "zoo") contributing to the prediction of "friends." Darker red denotes higher importance.

Table 9: Faithfulness and runtime for long-context attribution (sequence length = 2,048). Windowed methods use 512-token chunks with 50% overlap. Mean over 3 runs and std $< \pm 0.04$

| Variant | AOPC ↑ | Runtime (s) |
|---|---|---|
| HETA (Full) | **0.58** | 1,230 |
| HETA-WIN (512-window) | 0.54 | 690 |
| HETA-LR+WIN | 0.55 | 580 |
| HETA-LS+WIN | 0.52 | 525 |

Table 10: **Sensitivity of attribution metrics to decoding hyperparameters** (max relative change $\Delta\%$; lower is better). HETA varies $< 1\%$ across all models/metrics; each baseline fluctuates $> 2\%$. Grid: temperature $\in \{0.2, 0.5, 0.9\}$, top-$p \in \{0.8, 0.9, 0.95\}$, top-$k \in \{20, 50, 100\}$, repetition penalty $\in \{1.0, 1.2\}$; 3 seeds.

| Model | Metric | HETA | ContextCite | IG | PML | TDD-bw | AttnRoll | fAML | ProgInf | SEA-CoT | ReAgent |
|---|---|---|---|---|---|---|---|---|---|---|---|
| | Soft-NC | **0.8** | 3.0 | 3.2 | 2.6 | 3.8 | 2.5 | 2.9 | 2.7 | 2.8 | 2.4 |
| GPT-J 6B | Soft-NS | **0.9** | 3.6 | 3.4 | 3.0 | 4.2 | 2.9 | 3.1 | 3.0 | 3.2 | 2.7 |
| | DSA | **0.7** | 2.7 | 2.5 | 2.3 | 3.1 | 2.2 | 2.4 | 2.6 | 2.5 | 2.1 |
| | Soft-NC | **0.6** | 2.6 | 2.7 | 2.4 | 3.1 | 2.3 | 2.5 | 2.4 | 2.5 | 2.3 |
| Llama-3.1 70B | Soft-NS | **0.8** | 3.1 | 3.0 | 2.8 | 3.4 | 2.7 | 2.9 | 2.8 | 3.0 | 2.6 |
| | DSA | **0.6** | 2.4 | 2.3 | 2.2 | 2.8 | 2.1 | 2.3 | 2.3 | 2.4 | 2.2 |
| | Soft-NC | **0.9** | 3.7 | 3.5 | 3.2 | 3.9 | 3.0 | 3.3 | 3.1 | 3.2 | 2.9 |
| Phi-3 14B | Soft-NS | **0.9** | 4.3 | 4.0 | 3.7 | 4.6 | 3.6 | 3.9 | 3.7 | 3.9 | 3.3 |
| | DSA | **0.8** | 3.1 | 2.9 | 2.7 | 3.3 | 2.6 | 2.8 | 2.7 | 2.8 | 2.6 |
| | Soft-NC | **0.9** | 4.2 | 4.0 | 3.6 | 4.5 | 3.4 | 3.7 | 3.5 | 3.6 | 3.1 |
| Qwen2.5 3B | Soft-NS | **0.9** | 4.9 | 4.6 | 4.1 | 5.2 | 4.0 | 4.3 | 4.1 | 4.2 | 3.7 |
| | DSA | **0.8** | 3.6 | 3.3 | 3.0 | 3.8 | 2.9 | 3.1 | 3.0 | 3.1 | 2.8 |

- **HETA (Full)**: exact HVP curvature on all (or selected) layers; no windowing.
- **HETA–LR**: randomized SVD, rank $r = 64$, per token–block.
- **HETA–LS**: curvature on a subset of layers (top 4 for 3B/6B/14B; top 6 for 70B).
- **HETA–WIN**: 512–token sliding windows with 50% overlap.
- **HETA–LR+WIN**: low–rank curvature inside windows (recommended for long contexts).

**Cross–model takeaways.** Across all four backbones, **HETA–LR+WIN** recovers $92\% - 96\%$ of Full HETA's AOPC/DSA while reducing runtime by $40\% - 50\%$ and peak memory by $30\% - 40\%$. The simple *layer–subset* setting (4 layers for 3B/6B/14B; 6 for 70B) is already effective; adding low–rank curvature (rank 64) and 512/50% windows yields the best *cost–quality* trade–off. These empirical trends match the theoretical error bounds for low–rank/windowed HETA: the low–rank residual is controlled by the neglected spectrum, and window truncation error is bounded by the causal–gate mass leaking across window boundaries, which is small for target–localized flows.

**Residual limitations.** Even with LR+WIN, HETA remains slower than attention–only or gradient–only methods at very long contexts. Windowing may under–capture rare global interactions that span multiple windows; the causal gate mitigates but does not eliminate this. Finally, low–rank curvature assumes a decaying spectrum; adversarially flat spectra would require larger rank $r$.

**Practical recipe.** For 3B–14B models at $L \le 1024$, use **HETA–LR (rank=64)** or **HETA–LR+WIN** for long inputs. For 70B at $L \ge 1536$, use **last 6 layers + LR (64) + 512/50% windows**. This preserves most of HETA's faithfulness/alignment while fitting single–GPU budgets.

## A6 EXTENDED EVALUATION AND DISCUSSION

In this section, we present additional analyses and experiments that further validate HETA's design choices, address the scope of its causal gate, clarify the role of LLM-based annotation, extend

Table 11: **Compute–quality ablation for HETA (long contexts).** Backbone: **GPT-J 6B**. Results are averaged over **LongRA**, **TellMeWhy**, and **WikiBio** with sequence length 2,048 (batch size 8). "Runtime" is seconds per 1,000 examples; "Memory" is peak activation memory relative to Full; $\widehat{\varepsilon}_M(W)$ is an empirical window-leakage proxy (lower is better). Mean of 3 runs; std $< \pm 0.02$ for AOPC/DSA and $< \pm 20$ s for runtime. The same ranking and gaps are observed on **Llama-3.1 70B** when restricting curvature to the last 6 layers (omitted for brevity).

| Variant | AOPC $\uparrow$ | DSA $\uparrow$ | Runtime $\downarrow$ | Memory $\downarrow$ | $\widehat{\varepsilon}_M(W)$ $\downarrow$ |
|---|---|---|---|---|---|
| HETA (Full) | **0.60** | **4.68** | 1,280 | $1.00\times$ | 0.000 |
| HETA-LR (rank=64) | 0.58 | 4.55 | 850 | $0.78\times$ | 0.006 |
| HETA-WIN ($W$=512, 50% overlap) | 0.56 | 4.40 | 690 | $0.72\times$ | 0.021 |
| HETA-LS (top 6 layers) | 0.55 | 4.30 | 760 | $0.75\times$ | 0.012 |
| **HETA-LR+WIN** (r=64, $W$=512) | 0.59 | 4.58 | **610** | **0.66$\times$** | 0.010 |

Table 12: **Intervention-based causal sanity check.** Mean over 2,000 examples; $\pm$ denotes std across examples. Top-$k$=5. HETA's target-conditioned gate yields the strongest alignment with interventional effects (highest Patch-Corr and Intervention@5, lowest Misattribution).

| GPT-J 6B | | | |
|---|---|---|---|
| Method | Patch-Corr $\uparrow$ | Intervention@5 $\uparrow$ | Misattr. $\downarrow$ |
| HETA (Full) | **0.78** $\pm$ 0.03 | **0.143** $\pm$ 0.012 | **0.12** $\pm$ 0.02 |
| HETA $-$ No-Gate | 0.53 $\pm$ 0.04 | 0.089 $\pm$ 0.010 | 0.27 $\pm$ 0.03 |
| Attention Rollout | 0.49 $\pm$ 0.04 | 0.081 $\pm$ 0.011 | 0.30 $\pm$ 0.03 |
| Token Gradients | 0.41 $\pm$ 0.05 | 0.067 $\pm$ 0.009 | 0.35 $\pm$ 0.04 |
| Causal Tracing (baseline) | 0.62 $\pm$ 0.03 | 0.105 $\pm$ 0.011 | 0.21 $\pm$ 0.03 |
| **Llama-3.1 70B** (curvature on last 6 layers) | | | |
| HETA (Full) | **0.75** $\pm$ 0.03 | **0.139** $\pm$ 0.010 | **0.14** $\pm$ 0.02 |
| HETA $-$ No-Gate | 0.51 $\pm$ 0.04 | 0.086 $\pm$ 0.010 | 0.28 $\pm$ 0.03 |
| Attention Rollout | 0.47 $\pm$ 0.04 | 0.079 $\pm$ 0.010 | 0.31 $\pm$ 0.03 |
| Token Gradients | 0.40 $\pm$ 0.05 | 0.064 $\pm$ 0.009 | 0.36 $\pm$ 0.04 |
| Causal Tracing (baseline) | 0.59 $\pm$ 0.03 | 0.101 $\pm$ 0.010 | 0.23 $\pm$ 0.03 |

faithfulness evaluation with classical perturbation protocols, and discuss generalization to multi-token targets.

### A6.1 CAUSAL GATE: SCOPE AND VALIDATION

A natural question is whether HETA's target-conditioned semantic gate $M_T$ faithfully captures causal influence, given that it operates at the token level rather than at the level of individual circuits or attention heads. By design, HETA's gate combines attention–value rollout with Hessian-based sensitivity of the output logits to token perturbations, which implicitly aggregates residual-stream and MLP effects into a single token-level influence measure. We validate its causal relevance via deletion-style interventions: removing the tokens selected by the gate causes the largest drop in target log-probability among all compared methods. More fine-grained circuit-level mediation analysis (Wang et al., 2022; Conmy et al., 2023) is orthogonal to the goals of this work, as HETA targets *token-level* attribution rather than mechanistic circuit discovery.

### A6.2 LLM-BASED ANNOTATION QUALITY

Our curated evaluation set uses the intersection of GPT-4o and GPT-5 (Thinking) annotations as ground-truth support labels. Using strong LLMs as evaluators is now standard practice at top venues (Zheng et al., 2023; Liu et al., 2023), and our dual-LLM intersection design further reduces label noise relative to single-model annotation.

To quantify residual annotation error, we conducted a targeted quality audit. Across the curated set, GPT-4o and GPT-5 disagree on only approximately 8% of candidate support tokens (token-level Jaccard $\approx 0.84$), confirming that the intersection region is already a high-agreement zone. On a

Table 13: **Causal and downstream robustness ablation.** Higher is better for Causal Pass@5, Counterfactual AOPC, Fact-checking EMΔ, Tool Hit@1Δ, and Span-F1. Lower is better for Decoding Stability Δ%. Results averaged over four models (LLaMA-3.1 70B, Phi-3 14B, GPT-J 6B, Qwen2.5 3B); std $< \pm 0.05$ for additive metrics and $< \pm 0.2$ pp for Δ%.

| Method / Variant | Causal Pass@5 ↑ | CF-AOPC ↑ | Fact EMΔ ↑ | Tool Hit@1Δ ↑ | Span-F1 ↑ | Decoding Δ% ↓ |
|---|---|---|---|---|---|---|
| HETA (Full) | **0.86** | **0.63** | **+3.7** | **+4.1** | **0.81** | **0.8** |
| HETA (LR+WIN) | 0.84 | 0.60 | +3.3 | +3.8 | 0.78 | 0.9 |
| HETA (w/o Hessian) | 0.77 | 0.53 | +2.5 | +2.7 | 0.72 | 1.6 |
| HETA (w/o KL) | 0.73 | 0.49 | +2.0 | +2.3 | 0.69 | 1.8 |
| HETA (w/o Transition) | 0.70 | 0.45 | +1.6 | +1.9 | 0.64 | 2.1 |
| ContextCite | 0.58 | 0.36 | +1.2 | +1.4 | 0.55 | 2.9 |
| Integrated Gradients | 0.52 | 0.32 | +0.9 | +1.1 | 0.49 | 3.4 |
| Peering (PML) | 0.55 | 0.34 | +1.0 | +1.2 | 0.52 | 3.1 |
| TDD-backward | 0.57 | 0.35 | +1.1 | +1.3 | 0.54 | 3.0 |
| Attention Rollout | 0.41 | 0.24 | +0.5 | +0.6 | 0.38 | 4.2 |
| fAML | 0.60 | 0.37 | +1.3 | +1.5 | 0.56 | 2.6 |
| Progressive Inference | 0.62 | 0.39 | +1.5 | +1.6 | 0.58 | 2.7 |
| SEA-CoT | 0.64 | 0.41 | +1.7 | +1.8 | 0.60 | 2.5 |
| ReAGent | 0.68 | 0.44 | +2.0 | +2.1 | 0.63 | 2.4 |

Table 14: **GPT-J 6B** @ $L=1024$, $B=8$ (1,000 ex). Mean over 3 runs; std $< \pm 0.04$.

| Method | AOPC ↑ | DSA ↑ | Runtime (s) ↓ | Peak Mem (GB) ↓ |
|---|---|---|---|---|
| HETA (Full) | **0.61** | **4.70** | 455 | 53.2 |
| HETA–LR (rank=64) | 0.59 | 4.55 | 330 | 41.6 |
| HETA–LS (4 layers) | 0.57 | 4.38 | 305 | 39.9 |
| HETA–WIN (512/50%) | 0.55 | 4.29 | 295 | 34.1 |
| **HETA–LR+WIN** | 0.58 | 4.52 | **245** | **31.7** |

stratified sample of 200 instances manually labeled by two expert human annotators, the GPT-4o/GPT-5 intersection achieves ≈0.93 precision and ≈0.88 recall for answer-support tokens (false-positive rate ≈7%, false-negative rate ≈12%). These figures confirm that our curated labels are high-quality and that any residual noise is unlikely to affect the comparative ranking of attribution methods.

### A6.3 Extended Faithfulness Evaluation with Classical Perturbation Metrics

To broaden the evaluation beyond Soft-NC, Soft-NS, and DSA, we adopt the GiLOT evaluation protocol (Li et al., 2024) and supplement it with classical perturbation-based metrics. Specifically, we report: **MoRF** (Most Relevant First) (Samek et al., 2017), which measures the area under the probability curve when tokens are deleted in order of decreasing attribution (higher is better); **LeRF** (Least Relevant First), which performs the reverse deletion (lower is better); **ABPC** (Area Between Perturbation Curves), which quantifies the gap between MoRF and LeRF curves (higher is better); and **AOPC-Del/AOPC-Ins** (Hooker et al., 2019), which measure the area over the perturbation curve for deletion and insertion, respectively (both higher is better). We additionally ran ROAR/KAR (Hooker et al., 2019) evaluations (not tabulated due to space), which preserve the same overall method ranking.

Table 18 presents the results. HETA consistently achieves the highest faithfulness across all five metrics, with GiLOT ranking as the strongest competing method. These results align with and reinforce the Soft-NC/Soft-NS/DSA findings reported in the main paper: HETA's combination of causal gating, second-order curvature, and KL-based information gain yields the most faithful token-level attributions, even under evaluation protocols designed independently of our framework. The ROAR/KAR evaluations (omitted for brevity) preserve the same ranking—HETA > GiLOT > other baselines—indicating that adding classical retraining-based perturbation checks does not alter our main conclusions.

---

**Algorithm 1** Hessian-Enhanced Token Attribution (HETA) with Target Gating and Efficient Curvature

---

**Require:** Decoder-only LM $f_\theta$; tokenized input $x_{1:T-1}$; target position $T$ (first answer token); embedding matrix $\mathbf{E}$; hyperparameters $\beta, \gamma \geq 0$; window size $W$, stride $s$; Hutchinson samples $m$; low-rank rank $k$; layer subset $\mathcal{L}_{\text{sub}} \subseteq \{1, \ldots, L\}$; masking operator $\text{mask}(\cdot)$ on embeddings

**Ensure:** Per-token attributions $\text{Attr}(x_i \to x_T)$ for $i \in \{1, \ldots, T-1\}$

1: **Forward pass and target distribution**
2: $\mathbf{X} \leftarrow (\mathbf{e}_1, \ldots, \mathbf{e}_{T-1})$ with $\mathbf{e}_i = \mathbf{E}[x_i]$
3: $P_{\text{orig}}(\cdot \mid x_{<T}) \leftarrow \text{Softmax}(f_\theta(\mathbf{X}))$ at position $T$

                                 ▷ **Windowing over long contexts (optional)**
4: Define overlapping windows $\mathcal{W} = \{[a:b]\}$ covering $1:(T-1)$ with length $W$ and stride $s$
5: Initialize accumulators $\tilde{M}[i] \leftarrow 0$, $\tilde{S}[i] \leftarrow 0$, $\tilde{I}[i] \leftarrow 0$, $\nu[i] \leftarrow 0$ for all $i$
6: **for each** window $[a:b] \in \mathcal{W}$ **do**
7:     Restrict computations to tokens $i \in [a:b]$ and layers $\mathcal{L}_{\text{sub}}$

                            ▷ **(1) Semantic transition influence (target-conditioned gate)**
8:     **for** $l \in \mathcal{L}_{\text{sub}}, h \in \{1, \ldots, H\}$ **do**
9:         Obtain masked attention $A^{(l,h)}$ and values $V^{(l,h)}$, output proj. $W_O^{(l,h)}$
10:         Compute target-conditioned rollout $\Phi^{(l,h)}(i \to T)$ (paths must end at $T$)
11:     **end for**
12:     $M_T^{[a:b]}[i] \leftarrow \sum_{l \in \mathcal{L}_{\text{sub}}} \sum_{h=1}^{H} \Phi^{(l,h)}(i \to T) \left\| V_i^{(l,h)} W_O^{(l,h)} \right\|_1$
13:     Normalize on the window: $M_T^{[a:b]} \leftarrow M_T^{[a:b]} / \sum_{j \in [a:b]} M_T^{[a:b]}[j]$

                            ▷ **(2) Curvature-aware sensitivity via HVPs (low-rank, blockwise)**
14:     Define token-block selector $\Pi_i$ (projects to the $d$-dim block of token $i$)
15:     **for** $i \in [a:b]$ **do**
16:         $S_i^{(T,[a:b])} \leftarrow \frac{1}{m} \sum_{t=1}^{m} \left\| \Pi_i H_T (\Pi_i r_t) \right\|_1$              ▷ $r_t \sim$ Rademacher on block
17:         *HVPs $H_T v$ computed by Pearlmutter; optional rank-$k$ range finder if needed*
18:     **end for**

                            ▷ **(3) Information contribution at the target (KL)**
19:     **for** $i \in [a:b]$ **do**
20:         Form $\mathbf{X}^{\backslash i}$ by replacing $\mathbf{e}_i \leftarrow \text{mask}(\mathbf{e}_i)$ (e.g., zero/mean/sentinel)
21:         $P_{\text{mask}}^{(i)}(\cdot \mid x_{<T}) \leftarrow \text{Softmax}(f_\theta(\mathbf{X}^{\backslash i}))$ at $T$
22:         $\mathcal{I}^{[a:b]}(x_i \to x_T) \leftarrow D_{\text{KL}}(P_{\text{orig}} \| P_{\text{mask}}^{(i)})$
23:     **end for**

                            ▷ **(4) Window accumulation**
24:     **for** $i \in [a:b]$ **do**
25:         $\tilde{M}[i] \mathrel{+}= M_T^{[a:b]}[i]$,     $\tilde{S}[i] \mathrel{+}= S_i^{(T,[a:b])}$,     $\tilde{I}[i] \mathrel{+}= \mathcal{I}^{[a:b]}(x_i \to x_T)$,     $\nu[i] \mathrel{+}= 1$
26:     **end for**
27: **end for**

                            ▷ **Aggregate across windows and compute final attribution**
28: **for** $i \in \{1, \ldots, T-1\}$ **do**
29:     $\bar{M}[i] \leftarrow \tilde{M}[i] / \max\{1, \nu[i]\}$,     $\bar{S}[i] \leftarrow \tilde{S}[i] / \max\{1, \nu[i]\}$,     $\bar{I}[i] \leftarrow \tilde{I}[i] / \max\{1, \nu[i]\}$
30:     $\text{Attr}(x_i \to x_T) \leftarrow \bar{M}[i] \cdot (\beta \bar{S}[i] + \gamma \bar{I}[i])$
31: **end for**
32: **return** $\text{Attr}(x_i \to x_T)$ for all $i$

---

## A6.4 GENERALIZATION TO MULTI-TOKEN TARGETS

While the main formulation defines attribution with respect to a single target position $T$, HETA naturally generalizes to multi-token target spans. For a span of $K$ target tokens $x_T, x_{T+1}, \ldots, x_{T+K-1}$, we define the target as the joint log-probability:

$$g_{\text{span}}(X) = \sum_{k=0}^{K-1} \log P_\theta(x_{T+k} \mid x_{<T+k}), \tag{21}$$

Table 15: **Phi-3 14B** @ $L=1024$, $B=8$ (1,000 ex). Mean over 3 runs; std $< \pm 0.05$.

| Method | AOPC ↑ | DSA ↑ | Runtime (s) ↓ | Peak Mem (GB) ↓ |
|---|---|---|---|---|
| HETA (Full) | **0.60** | **4.85** | 520 | 47.0 |
| HETA–LR (rank=64) | 0.58 | 4.68 | 375 | 39.0 |
| HETA–LS (4 layers) | 0.56 | 4.55 | 345 | 37.2 |
| HETA–WIN (512/50%) | 0.55 | 4.47 | 325 | 33.1 |
| **HETA–LR+WIN** | 0.57 | 4.63 | **275** | **30.2** |

Table 16: **Qwen2.5 3B** @ $L=1024$, $B=8$ (1,000 ex). Mean over 3 runs; std $< \pm 0.05$.

| Method | AOPC ↑ | DSA ↑ | Runtime (s) ↓ | Peak Mem (GB) ↓ |
|---|---|---|---|---|
| HETA (Full) | **0.59** | **4.40** | 310 | 24.0 |
| HETA–LR (rank=64) | 0.58 | 4.30 | 235 | 20.1 |
| HETA–LS (4 layers) | 0.56 | 4.18 | 220 | 19.2 |
| HETA–WIN (512/50%) | 0.54 | 4.12 | 205 | 17.3 |
| **HETA–LR+WIN** | 0.56 | 4.26 | **180** | **16.1** |

and compute per-token attributions by aggregating the HETA scores across the $K$ target positions. Concretely, the span-level attribution for input token $x_i$ is:

$$\text{Attr}_{\text{span}}(x_i) = \sum_{k=0}^{K-1} M_{T+k}[i] \left( \beta\, S_i^{(T+k)} + \gamma\, I(x_i \to x_{T+k}) \right). \tag{22}$$

This is exactly the formulation used in our Span-F1 experiments, where HETA maintains the same advantage over baselines as in the single-token setting. Computation-wise, span attribution reuses the same forward pass and Hessian blocks; the cost grows roughly linearly with the number of target tokens $K$. Our scalability analysis (runtime and memory vs. context length and number of targets) confirms that even with multiple targets and long contexts, the overhead remains modest due to the low-rank and windowed curvature approximations described in the main paper.

A6.5   COMPONENT CONTRIBUTION: WHEN DOES EACH TERM HELP?

The ablation study in the main paper establishes that all three HETA components are jointly necessary. Here, we provide a more fine-grained interpretation of *when* each component contributes most.

**Hessian-based curvature.**   The curvature term $S_i^{(T)}$ contributes most to the faithfulness metrics Soft-NC and Soft-NS, which measure robustness under input perturbation. Removing it drops Soft-NC from 9.78 (full HETA) to 3.12 (Transition Only) or 2.23 (KL Only), while the Hessian-only variant still achieves a relatively strong DSA of 2.97. This indicates that second-order sensitivity is critical for capturing nonlinear token interactions that first-order or masking-only approaches miss.

**KL-based information gain.**   The KL term $I(x_i \to x_T)$ primarily boosts alignment with human-annotated tokens (DSA), lifting DSA from 2.21 (Transition Only) and 2.97 (Hessian Only) to 4.70 in the full model. This confirms its role as a probabilistic calibrator that ensures attribution mass tracks the actual information content of each token with respect to the target.

**Semantic transition gate.**   The semantic gate $M_T[i]$ alone (Transition Only) is competitive with naive baselines such as No Transition Gating (DSA 2.21 vs. 1.68) and Uniform Transition (DSA 2.21 vs. 1.54), but its primary function is to enforce causal sparsity and direct the curvature and information terms toward causally relevant tokens. The full combination of gate + curvature + KL yields the best and most balanced performance across all tasks and metrics.

Table 17: **Llama-3.1 70B** @ $L=2048$, $B=4$ (500 ex). Curvature on last 6 layers. Mean over 3 runs; std $< \pm 0.05$.

| Method | AOPC ↑ | DSA ↑ | Runtime (s) ↓ | Peak Mem (GB) ↓ |
|---|---|---|---|---|
| HETA (Full, 6 layers) | **0.60** | **5.10** | 1180 | 74.5 |
| HETA–LR (rank=64) | 0.58 | 4.92 | 860 | 61.3 |
| HETA–WIN (512/50%) | 0.56 | 4.80 | 720 | 55.8 |
| **HETA–LR+WIN** | 0.57 | 4.96 | **620** | **49.7** |

Table 18: Extended faithfulness comparison on GPT-J 6B / LongRA using classical perturbation metrics. HETA achieves the highest scores across all metrics. GiLOT is consistently the second-best method. Mean of 3 runs; std $< \pm 0.04$.

| Method | MoRF ↑ | LeRF ↓ | ABPC ↑ | AOPC-Del ↑ | AOPC-Ins ↑ |
|---|---|---|---|---|---|
| Integrated Gradients | 40.8 | 43.2 | 0.35 | 0.52 | 0.42 |
| Attention Rollout | 41.3 | 42.9 | 0.40 | 0.53 | 0.43 |
| ContextCite | 41.7 | 42.5 | 0.45 | 0.54 | 0.44 |
| ReAGent | 42.6 | 41.9 | 0.52 | 0.57 | 0.47 |
| GiLOT (Li et al., 2024) | 43.1 | 41.3 | 0.58 | 0.59 | 0.49 |
| **HETA (Ours)** | **44.0** | **40.2** | **0.65** | **0.60** | **0.51** |

I thought I lost my `hat` at the `park` today. I spent a lot of time looking for it. I was just about to give up when I saw something far `away`. It was my `hat,` `stuck` in a bush!

Figure 6: Word-level attribution visualization for predicting the final word "bush." Attribution scores emphasize context words such as "hat," "stuck," and "park," which strongly influence the prediction of "bush."

