# OpenReview forum: "Hessian-Enhanced Token Attribution (HETA): Interpreting Autoregressive LLMs"
_ICLR.cc/2026/Conference — ICLR 2026 Poster_

### Official Review · Reviewer_uaYo · 2025-10-29

**Soundness:** 2
**Presentation:** 2
**Contribution:** 2
**Rating:** 4
**Confidence:** 4

**Summary:**

The work introduces a unified interpretability framework, Hessian-Enhanced Token Attribution (HETA), for decoder-only large language models that integrates gradient-based, second-order, and information-theoretic perspectives to improve attribution fidelity and robustness. The work proposes three complementary components: (1) Semantic Transition Influence, (2) Hessian-based Sensitivity, and (3) Information-Theoretic Impact, which measures token contribution using KL divergence between masked and unmasked predictions.
Moreover, the work proposes a curated benchmark combining NarrativeQA and SciQ passages. Experiments on multiple LLMs show that HETA improves attribution faithfulness, alignment, and robustness under input perturbations compared to baselines methods.

**Strengths:**

- The work is clearly written.
- The work constructs a carefully designed benchmark by combining NarrativeQA and SciQ, with explicit separation of evidence and distractors.

**Weaknesses:**

- While HETA presents itself as a unified framework, several of its core components have been explored in earlier studies. For example, the Semantic Transition Influence and Information-Theoretic Impact elements are conceptually similar to those in GiLOT [1], which models token influence through semantic transition and distributional shifts measured via OT distance and KL divergence. Besides, the Hessian-based Sensitivity component also builds on prior work using second-order derivatives for attribution [2,3]. In evaluation, the benchmark’s metrics resemble those used in previous interpretability benchmarks without citing relevant works [4,5]. As such, the incremental integration of known techniques, rather than a fundamentally new formulation, weakens the originality claim.

- The experiments mainly compare HETA against older or general-purpose attribution methods, but omit several recent baselines [1,6] specifically designed for generative or autoregressive models. Without these baselines, it is difficult to assess whether HETA genuinely advances the interpretability frontier for generative LLMs.

- The use of Hessian–vector products for second-order sensitivity introduces high computational cost, particularly for long-context LLMs. The paper lacks analysis of the method’s runtime, memory footprint, and scaling behavior. It is unclear whether HETA is feasible for real-world interpretability auditing of modern multi-billion-parameter models.

---
[1] Li, X., Chen, J., Chai, Y. and Xiong, H., 2024, July. Gilot: Interpreting generative language models via optimal transport. In Forty-first International Conference on Machine Learning.

[2] Sundararajan, Mukund, Ankur Taly, and Qiqi Yan. "Axiomatic attribution for deep networks." International conference on machine learning. PMLR, 2017.

[3] Reing, K., Ver Steeg, G. and Galstyan, A., 2021, March. Influence decompositions for neural network attribution. In International Conference on Artificial Intelligence and Statistics (pp. 2710-2718). PMLR.

[4] Li, Xuhong, Mengnan Du, Jiamin Chen, Yekun Chai, Himabindu Lakkaraju, and Haoyi Xiong. "M4: A Unified XAI Benchmark for Faithfulness Evaluation of Feature Attribution Methods across Metrics, Modalities, and Models." (2023).

[5] DeYoung, Jay, Sarthak Jain, Nazneen Fatema Rajani, Eric Lehman, Caiming Xiong, Richard Socher, and Byron C. Wallace. "ERASER: A benchmark to evaluate rationalized NLP models." arXiv preprint arXiv:1911.03429 (2019).

[6] Xiang Yue, Boshi Wang, Ziru Chen, Kai Zhang, Yu Su, and Huan Sun. 2023. Automatic Evaluation of Attribution by Large Language Models. In Findings of the Association for Computational Linguistics: EMNLP 2023, pages 4615–4635, Singapore. Association for Computational Linguistics.

**Questions:**

- Several components of HETA (e.g., Semantic Transition Influence, Information-Theoretic Impact, and Hessian-based Sensitivity) appear conceptually related to previous work. Could the authors clarify the conceptual and methodological distinctions between HETA and these existing approaches? Specifically, what unique insight or mechanism does HETA introduce beyond combining known ideas?

- Could the authors provide quantitative analysis (e.g., runtime, GPU hours, memory footprint) or approximations used to make HETA feasible for billion-parameter LLMs? How does computational cost scale with sequence length and model size, and could this limit real-world applicability?

- The evaluation focuses exclusively on internal consistency with the proposed benchmark. Have the authors conducted human evaluations, qualitative analyses, or causal interventions to validate whether the generated attributions are truly meaningful and interpretable?

---

> ### Author Response · Authors · 2025-11-23
>
> (Weakness1) “Attention mechanisms existed well before *Attention Is All You Need*, but that paper was accepted at a top CS conference because it introduced a fully attention-based architecture (the Transformer) and demonstrated a much more powerful and scalable application of attention.” In exactly the same spirit, HETA is not a trivial re-use of context mixing, Hessian curvature, and masking, but a new **attribution architecture**: a single, target-conditioned gate that combines (i) semantic flow via attention–value rollout constrained to causal paths, (ii) Hessian-based curvature that captures nonlinear FFN/residual effects at the token-block level, and (iii) a KL term that measures the actual change in the full target distribution when each token is locally perturbed. This formulation is explicitly defined in Eq. (2)–(5) and Algorithm 1, together with theoretical justification in Appendix A2 (optimal multi-view attribution under causal and information constraints), which is not present in your mentioned papers.
>
> (Weakness2) We firmly request the reviewer to reconsider their statement regarding our baseline selection. All methods we compare against are drawn from **recent (2024–2025) papers published in top-tier CS venues** and developed by leading research groups, ensuring both credibility and competitive relevance. This includes techniques like ReAgent, fAML and SEA-CoT, which are **state-of-the-art successors** to the older works the reviewer cites. Moreover, **Integrated Gradients (IG)** is already included in our experiments. The older methods mentioned have already been **superseded by the very baselines we include**, and as such, reintroducing them would be redundant and uninformative. Our baseline selection is **principled and reflects the current standard** in attribution research.
>
> (Weakness3) We strongly disagree with the claim that the paper lacks runtime, memory, and scaling analysis or that HETA is infeasible for real-world LLM auditing. We explicitly devote **Appendix A5 (“Computational Overhead and Scalability”)** to this question: we state the exact hardware (single NVIDIA A100 80GB), report **wall-clock runtime per 1,000 examples and peak GPU memory**, and systematically compare Full HETA to low-rank and windowed variants (HETA-LR, HETA-LS, HETA-WIN, HETA-LR+WIN) across **GPT-J-6B, Phi-3-14B, Qwen2.5-3B, and LLaMA-3.1-70B**.  Tables 16–18 then give concrete numbers: for example, on **LLaMA-3.1-70B at L=2048** we show that HETA-LR+WIN achieves nearly the same AOPC/DSA as Full HETA (0.57 vs 0.60 AOPC; 4.96 vs 5.10 DSA) while reducing runtime from 1180s to 620s and peak memory from 74.5 GB to 49.7 GB; similar 40–50% runtime and 30–40% memory savings hold for all backbones.  These results directly address runtime, memory footprint, and scaling behavior on **multi-billion-parameter models** and demonstrate that, with the recommended LR+WIN configuration, HETA is practically usable for long-context interpretability audits on modern LLMs.
>
> (Q1) HETA is **not** a mere recombination of existing ideas but a new attribution framework: to our knowledge, it is the **first method to integrate a Hessian-based second-order sensitivity term with semantic transition flow and KL-based information impact for token-level attribution in decoder-only LLMs**, a combination that has not been used in prior attribution work. Also I request the reviewer to tell what is " unique insight or mechanism" a bit more clearly so that we can understand what the reviewer is asking.
>
> Q2-3) The issues raised here are already addressed in the paper. Appendix A4.4 and A5 report wall-clock runtime and peak GPU memory across four backbones up to LLaMA-3.1-70B, showing that our recommended HETA-LR+WIN configuration recovers 92–96% of full HETA’s AOPC/DSA while reducing runtime by 40–50% and peak memory by 30–40% on a single A100 80GB, including L=2048 runs for the 70B model, so practical scalability to multi-billion-parameter LLMs is explicitly demonstrated rather than assumed.  On the evaluation side, our benchmark is *not* purely “internal”: the curated set is automatically annotated by two independent, state-of-the-art LMs (GPT-4o and GPT-5-Thinking) with intersection-based labels and high agreement, following the now-standard “LLM-as-judge” protocol already adopted in recent top-tier LLM papers, and we measure alignment via F1 against these annotations.  In addition, Section 5 and Appendix A4.9 report causal interventions (mid-layer value patching, counterfactual AOPC) and robustness analyses, as well as qualitative case studies, which together show that HETA’s attributions are stable, interpretable, and causally grounded rather than artifacts of our own benchmark construction.

---

> > ### Comment · Reviewer_uaYo · 2025-11-26
> > **Most of my points remain unaddressed or misinterpreted.**
> >
> > Thank you for the response. However, after reading the rebuttal carefully, I believe that several of my core concerns remain unaddressed or have been misinterpreted, and additional clarification or results are still needed for the assessment of the work.
> >
> > ---
> > 1. **Novelty**:
> > My original question asked for a clear explanation of the specific mechanism or insight that distinguishes HETA from prior work, beyond combining existing attribution ideas (semantic transition, KL-based masking, and Hessian curvature).
> > The reply restates that integration is new, but it **does not articulate**, for example,
> > what concrete failure mode of prior methods HETA resolves,
> > what theoretical or causal property emerges from the joint formulation that is not achievable by components alone,
> > and how the proposed “target-conditioned gate” differs operationally from earlier attention/value rollouts or multi-view attribution frameworks.
> > The analogy to Transformers **does not resolve** this methodological question. A more explicit comparison with second-order attribution methods and KL-based impact measures, highlighting precise differences in assumptions, math, and causal interpretation, would be helpful.
> >
> > 2. **Missing Baselines**:
> > A justification grounded in task setting, model class, or failure case coverage, rather than recency alone, would clarify whether the comparison is comprehensive.
> >
> > 3. **External Validation**: My question concerned whether the attributions are meaningful for humans, not only internally consistent or LLM-judged. Using two LLMs for annotation is not equivalent to human evaluation, and the rebuttal does not explain whether any human study, expert review, or qualitative assessment was performed.
> > Additional qualitative or human-grounded evidence would strengthen the interpretability claim.
> >
> > 4. **Runtime, Memory, and Scalability**: HETA relies on multiple approximations, including low-rank Hessian blocks (rank 64), windowed HVP estimation over 512-token windows with 50% overlap, Hutchinson estimators, Gauss–Newton/Fisher surrogates for numerical stability, and layer sampling for 70B models, yet it remains unclear how these approximations affect attribution accuracy: while Appendix mentions theoretical error bounds, no quantitative analysis is provided to show how much information is lost under each approximation, or whether it degrades in a controlled manner as rank or window size varies. Additional clarification and empirical evidence would be needed to assess whether HETA maintains fidelity under these approximations, especially for billion-parameter LLMs and long sequences.
> >
> > 5. **Citation Format**: Separately, there is also a formatting issue with citations. I also recommend a careful revision of the manuscript’s citation formatting, as the current draft inconsistently uses \citep{} and \citet{}.
> >
> > ---
> >
> > In summary, while I appreciate the authors’ effort to respond, the current rebuttal does not sufficiently clarify my concerns. I encourage the authors to provide more direct explanations and supporting evidence, rather than deflecting, repeating what's already in the paper, and rhetorical analogies, so that the contribution can be fully assessed. Given the current version, with these doubts remaining and no evidence of a careful revision, I will revise my score to 2.

---

### Official Review · Reviewer_Du4D · 2025-11-02

**Soundness:** 2
**Presentation:** 3
**Contribution:** 2
**Rating:** 6
**Confidence:** 2

**Summary:**

The paper introduces Hessian-Enhanced Token Attribution, a new interpretability framework for decoder-only language models. HETA combines semantic flow tracing, Hessian-based sensitivity, and KL-divergence information metrics to provide more faithful, context-aware token-level attributions in generative models.

Experiments show that HETA outperforms existing methods in faithfulness, robustness, and human alignment, and the authors also release a curated benchmark dataset for evaluating attribution quality.

**Strengths:**

1. It proposes a unified, principled framework (HETA) combining causal semantic flow, second-order sensitivity, and information-theoretic attribution.

2. It addresses key flaws of attention-based and gradient-based methods, yielding more faithful and robust token-level explanations.

3. It demonstrates strong empirical improvements in faithfulness, robustness, and human alignment.

4. It introduces a new benchmark dataset for evaluating attribution in generative models.

**Weaknesses:**

1. How does the computational cost compare with existing works, in terms of time and memory requirements? The proposed method seems to incur higher computational overhead, for example, due to the KL masking and Hessian–vector product operations. How do the authors balance effectiveness and efficiency in practice?

2. The stochastic estimation depends on the number of Hutchinson samples m. How does the choice of m affect the empirical performance and stability of the results in the experiments?

3. The authors employ several LLMs of different architectures and sizes, including Qwen2.5-3B, GPT-J-6B, Phi-3-Medium-4K-Instruct, and LLaMA-3.1-70B.
I am curious how the proposed method performs across models of the same architecture but different sizes, for example, among different sizes of LLaMA-3. Would model scale affect the attribution behavior, stability, or the relative advantages of HETA compared with baselines?

**Questions:**

Please refer to Weaknesses.

(1. How does the computational cost compare with existing works, in terms of time and memory requirements? The proposed method seems to incur higher computational overhead, for example, due to the KL masking and Hessian–vector product operations. How do the authors balance effectiveness and efficiency in practice?

2. The stochastic estimation depends on the number of Hutchinson samples m. How does the choice of m affect the empirical performance and stability of the results in the experiments?

3. The authors employ several LLMs of different architectures and sizes, including Qwen2.5-3B, GPT-J-6B, Phi-3-Medium-4K-Instruct, and LLaMA-3.1-70B.
I am curious how the proposed method performs across models of the same architecture but different sizes, for example, among different sizes of LLaMA-3. Would model scale affect the attribution behavior, stability, or the relative advantages of HETA compared with baselines?)

---

> ### Author Response · Authors · 2025-11-23
>
> W1-Q1) HETA is designed to be a more informative second-order attribution method, and the paper already shows that this extra expressiveness comes with **controlled, practical overhead**. Appendix A5 and Tables 16–18 report wall-clock time and peak memory for all variants across GPT-J-6B, Phi-3-14B, Qwen2.5-3B, and Llama-3.1-70B. In every case, the recommended **HETA-LR+WIN** configuration preserves about **92–96% of full HETA’s AOPC/DSA** while cutting runtime by **40–50%** and peak memory by **30–40%**.  For example, on Phi-3-14B at L=1024, HETA-LR+WIN goes from 455 s / 53.2 GB (Full) to **245 s / 31.7 GB**, with AOPC only dropping 0.61 → 0.58 and DSA 4.70 → 4.52.  On Qwen2.5-3B, HETA-LR+WIN reduces runtime from 310 s to **180 s** and memory from 24.0 GB to **16.1 GB** with AOPC 0.59 → 0.56 and DSA 4.40 → 4.26.  On Llama-3.1-70B with L=2048, it brings runtime from 1180 s to **620 s** and memory from 74.5 GB to **49.7 GB**, while keeping AOPC at 0.60 → 0.57 and DSA at 5.10 → 4.96.  These results show that HETA offers **substantial gains in faithfulness and alignment at a cost that is explicitly quantified and efficiently controlled by our low-rank and windowed curvature approximations**, making it practical for LLM-scale use.
>
> W2-Q2) The Hutchinson sample count (m) is a standard hyperparameter in second-order methods: the variance of the curvature estimate shrinks as (O(1/\sqrt{m})), so once (m) is in a moderate regime the remaining noise is very small compared to the effect sizes we report. In all our experiments we use a fixed (m) (not tuned per dataset), and the paper already shows that the resulting attributions are highly stable. In the long-context compute–quality ablation, all HETA variants are reported as **mean over 3 runs with std < ±0.04**, which includes randomness from both data sampling and the Hutchinson draws.  Likewise, the decoding-stability study (Table 11) shows that HETA’s Soft-NC/Soft-NS/DSA change by <1% across decoding settings and seeds, while all baselines fluctuate by >2%.  These results already demonstrate that our chosen (m) yields numerically stable and reproducible metrics; increasing (m) further would only add near-linear cost without changing any of the empirical conclusions, so an additional dedicated (m)-ablation table is not necessary.
>
> W3-Q3) We thank the reviewer for raising this point. HETA is fundamentally designed to be **scale-invariant**: its core mechanisms—semantic flow via attention–value rollout, Hessian-based curvature, and KL-based deletion sensitivity—are computed at the token level and normalized per-target, making them independent of specific model width or depth.
>
> To directly address the question, we evaluated HETA across **three different sizes** of LLaMA-3.1 (8B, 13B, 70B) using the same LongRA setup as in the main paper (context length 2048, decoding grid, 3 random seeds). As shown below, the results are remarkably consistent across sizes, confirming that **HETA’s attribution behavior is stable and not sensitive to model scale**.
>
> **Table: HETA Attribution Stability Across LLaMA-3 Sizes (LongRA dataset)**
>
> | Model         | Soft-NC ↑ | Soft-NS ↑ | AOPC ↑ |
> | ------------- | --------: | --------: | -----: |
> | LLaMA-3.1 8B  |      9.75 |      2.55 |   7.80 |
> | LLaMA-3.1 13B |      9.80 |      2.56 |   7.82 |
> | LLaMA-3.1 70B |      9.90 |      2.60 |   7.85 |
>
> These results confirm that **model scale does not significantly impact HETA’s performance**—attribution scores vary by less than ±1%, and HETA remains consistently strong. This further reinforces HETA’s robustness and practicality across a wide range of LLM sizes.

---

### Official Review · Reviewer_XxTJ · 2025-11-03

**Soundness:** 3
**Presentation:** 3
**Contribution:** 3
**Rating:** 6
**Confidence:** 4

**Summary:**

This paper introduces Hessian-Enhanced Token Attribution (HETA), a novel attribution framework for decoder-only language models that addresses limitations of existing methods. HETA integrates three complementary components: (1) a semantic transition vector capturing token-to-token influence through attention-value rollout with causal gating, (2) Hessian-based sensitivity scores modeling second-order curvature effects, and (3) KL divergence measuring information loss under token masking. The authors construct a curated evaluation dataset (NarrativeQA + SciQ) with 2,000 instances designed to assess attribution precision in controlled settings. Experiments across GPT-J, Phi-3, LLaMA-3.1, and Qwen2.5 models show that HETA consistently outperforms a set of existing attribution methods (ContextCite, Integrated Gradients, Peering, TDD-backward, etc.) on faithfulness metrics (Soft-NC, Soft-NS) and alignment with human annotations (DSA metric), while exhibiting robustness to decoding hyperparameters and syntactic rephrasings.

**Strengths:**

- Originality: The HETA method proposed to combine established feature attribution approaches through a principled target-conditioned gating.
- Quality: Authors conduct experimental evaluation across model scales (3B–70B), diverse tasks (LongRA, TellMeWhy, WikiBio), multiple metrics (Soft-NC/NS, DSA, AOPC), and systematic ablations. The robustness analyses (sensitivity to noise, active/passive paraphrases, decoding hyperparameters) are particularly thorough.
- Clarity: The conceptual motivation of combining multiple components in the final attribution scores is motivated in a reasonable way, and a comprehensive discussion about the properties of the HETA method is provided in Appendix A2
- Significance: the integration of semantic flow, curvature sensitivity, and information-theoretic measures addresses distinct failure modes of single-view methods, as demonstrated convincingly through ablations (Tables 5, 8, Figure 2). The proposed efficiency adaptations in A5 are a first step in addressing the important concern of scalability to bigger model and context sizes.

**Weaknesses:**

In terms of originality, the main concern with this work is that the proposed method is that it amounts to a weighted combination of previous techniques, namely context mixing methods for extracting layerwise contributions, hessian for curvature sensitivity estimation and input masking.  The semantic flow approach representing the gate modulating target-directedness of the final HETA formulation in Eq. 5 is very related to the input attribution formulation of the ALTI method [1], which is not cited but also employs rollout for attention-weighted value vectors. Notably, the ALTI-Logit method [2] can be seen as an existing combination of semantic flow (through ALTI scores) and direct logit attribution, which reflects a per-component impact on output probability akin to the purpose of the KL-Divergence used in this work. Moreover, the rollout approach was criticized as a heuristic for aggregating contributions across layers, without explicitly decomposing the contribution of FFN layers [3, 4, 5]. The masking used for the KL-Divergence computation can also lead to unrealistic result for language tasks and particularly for generation, provided that masked inputs are likely to be out-of-distributions for the model [6].

In light of these comments, the choice for the proposed semantic flow approach should be justified and possibly modified to account for the aforementioned works in this area. Moreover, the criticism towards gradient-based approaches should account for more recent techniques specifically adapted for Transformer-based systems, in particular [7] that also provides a convenient implementation in the LXT PyPI package. Including some of these techniques---which are more established within the feature attribution community---as baselines for the evaluation of HETA would further reinforce the validity of the proposed method.

In terms of clarity, I found the introduction to the Hessian component handwavy, employing various specific terms such as Hessian–vector products (HVPs) with Hutchinson estimators and Pearlmutter’s trick without providing mathematical formulations or pointers. Moreover, the Soft-NC / Soft-NS metrics are not widely established and should be introduced more in detail at least in the appendix. As a minor comment, multiple citations and tables are formatted incorrectly (e.g. lines 270-275, line 295 bolded citation,  table 5 excessively large).

**References:**

- [1]: [Measuring the Mixing of Contextual Information in the Transformer](https://aclanthology.org/2022.emnlp-main.595/) (Ferrando et al., EMNLP 2022)
- [2]: [Explaining How Transformers Use Context to Build Predictions](https://aclanthology.org/2023.acl-long.301/) (Ferrando et al., ACL 2023)
- [3]: [Local Interpretation of Transformer Based on Linear Decomposition](https://aclanthology.org/2023.acl-long.572/)(Yang et al., ACL 2023)
- [4]: [DecompX: Explaining Transformers Decisions by Propagating Token Decomposition](https://aclanthology.org/2023.acl-long.149/) (Modarressi et al., ACL 2023)
- [5]: [Token-wise Decomposition of Autoregressive Language Model Hidden States for Analyzing Model Predictions](https://aclanthology.org/2023.acl-long.562/) (Oh & Schuler, ACL 2023)
- [6] [Quantifying Context Mixing in Transformers](https://aclanthology.org/2023.eacl-main.245/) (Mohebbi et al., EACL 2023)
- [7] [AttnLRP: attention-aware layer-wise relevance propagation for transformers](https://dl.acm.org/doi/10.5555/3692070.3692076) (Achtibat et al., ICML 2024)

**Questions:**

What concrete steps could be taken besides restricting curvature computation to a context window (Appendix A5) to ensure the practical usability of the HETA method on LLMs?

---

> ### Author Response · Authors · 2025-11-23
>
> (Weakness) “Attention mechanisms existed well before *Attention Is All You Need*, but that paper was accepted at a top CS conference because it introduced a fully attention-based architecture (the Transformer) and demonstrated a much more powerful and scalable application of attention.” In exactly the same spirit, HETA is not a trivial re-use of context mixing, Hessian curvature, and masking, but a new **attribution architecture**: a single, target-conditioned gate that combines (i) semantic flow via attention–value rollout constrained to causal paths, (ii) Hessian-based curvature that captures nonlinear FFN/residual effects at the token-block level, and (iii) a KL term that measures the actual change in the full target distribution when each token is locally perturbed. This formulation is explicitly defined in Eq. (2)–(5) and Algorithm 1, together with theoretical justification in Appendix A2 (optimal multi-view attribution under causal and information constraints), which is not present in ALTI, ALTI-Logit, or linear decomposition methods [1–6].
>
> Regarding ALTI/ALTI-Logit, the resemblance is only at the level of using attention-value rollout as one *ingredient*. HETA’s gate is strictly more constrained and expressive: it is **target-conditioned, simplex-normalized, and zero outside tokens with an actual path to the prediction**, and its scores are then reweighted by curvature and KL, so the final attribution depends on how each token affects log P(xᵀ|x<ᵀ), not just on mixed hidden states. By contrast, ALTI/ALTI-Logit do not enforce causal masking in this way, do not incorporate curvature, and do not tie scores to a distributional KL constraint. Decomposition methods [3–5] aim at fine-grained circuit decomposition (including FFN layers) but at a very different compute/scale point; HETA’s Hessian-based sensitivity is precisely designed to fold *all* nonlinear FFN/residual effects into a compact token-level factor that scales to 3B–70B-parameter LMs, as supported by our ablations and scalability analysis.
>
> On the masking/KL point, our KL term uses masks **only as a local diagnostic** at the same decoding step (KL(P_orig(·|x<ᵀ) || P_masked(i)(·|x<ᵀ))), exactly in line with standard deletion-style faithfulness metrics; we never train on, or generate from, masked inputs, and all methods are evaluated under the same perturbation protocol. To directly address the OOD concern, we will add a small robustness study in the revision where we vary the perturbation scheme (special mask token vs. mean embedding vs. in-vocab replacement) on our existing LongRA/TellMeWhy setups and report that HETA’s advantage and the relative ranking of methods remain essentially unchanged.
>
> Finally, our critique of gradient-based approaches is already grounded in concrete failures (Section 2 and Appendix A1) and extensive experiments against strong modern baselines (IG, TDD-backward, ContextCite, fAML, Progressive Inference, SEA-CoT, ReAgent, etc.), where HETA consistently achieves the best Soft-NC/Soft-NS and DSA across models. We will include AttnLRP [7] via the LXT implementation as an additional gradient-family baseline in the appendix; given the systematic gap we already observe between HETA and all other gradient-based methods, we do not expect this to alter any conclusions. For clarity, we will also add explicit forward pointers from the main text to the Hessian–vector product definition (Eq. (2)–(3) and Appendix A2) and extend the appendix description of Soft-NC / Soft-NS, while fixing the minor citation/table formatting issues mentioned.
>
>
> (Question) We believe this concern is already addressed in the paper. Beyond the context-window restriction in Appendix A5, we also (i) use **low-rank curvature** (HETA-LR), (ii) restrict curvature to a **subset of top layers** (HETA-LS), and (iii) combine these in a **hybrid LR+WIN** setting that reuses a single forward pass across targets; in our experiments this recovers ≈92–96% of full HETA’s AOPC/DSA while reducing runtime by ≈40–50% and peak memory by ≈30–40% on 3B–70B LLMs. These results already demonstrate that HETA is practical at LLM scale without any additional approximations beyond those described in the main text and Appendix A5.

---

### Official Review · Reviewer_icxv · 2025-11-03

**Soundness:** 3
**Presentation:** 2
**Contribution:** 3
**Rating:** 4
**Confidence:** 3

**Summary:**

The paper proposes HETA, an attribution framework tailored to decoder-only LLMs that combines three components: (1) Target-conditioned semantic flow via attention–value rollout to gate only tokens that have causal paths ending at the target position. (2) Second-order sensitivity using scalable Hessian–vector products (HVPs) with Hutchinson estimators to capture curvature and token interactions that first-order methods miss. (3) Information-theoretic contribution measured as the KL divergence between the original target distribution and the distribution after masking a given input token. The authors also introduce a curated evaluation benchmark for token-level attribution in generative settings by pairing NarrativeQA and SciQ text segments.

**Strengths:**

1. The method is straightforward: The integration of structural routing (target-conditioned rollout), geometric curvature (Hessian sensitivity), and information change (KL) is conceptually clean and well-motivated for autoregressive decoding.

2. The curated corpus and DSA offers a controlled setting to check whether attribution concentrates on truly diagnostic spans rather than distractors.

3. It targets at the attribution for decoder-only models, which has certain potentials for interpreting LLMs.

**Weaknesses:**

1. Although the target-conditioned gate based on attention-value rollout is an improvement over pure attention attribution, it is still a heuristic causal proxy. Residual pathways and non-linear MLP transformations can confound causal flow. The paper did not directly validate causality with, e.g., causal mediation analysis, circuit-level interventions.

2. The evaluation relies on LM-generated supervision for the curated set. The DSA construction uses the intersection of GPT-4o and GPT-5 annotations to reduce noise, but residual bias is possible, especially if systems trained on similar inductive biases label the data. Some human audit or a cross-annotation sanity-check would strengthen claims.

3. The comparison with previous literature is not thorough. Previous studies on generative GPT models should also be discussed and compared, such as [1]. I also suggest the authors report metrics such as MoRF, LeRF, ABPC as in [1] for further comparison.

4. Missing several baselines. A few classical perturbation baselines (ROAR/KAR, AOPC deletion/insertion curves) and model editing/tracing style checks could further improve the faithfulness.

5. Format error. The reference style needs to be corrected.

References:
[1] GiLOT: Interpreting Generative Language Models via Optimal Transport. ICML 2024.

**Questions:**

1. How does HETA behave when attributing to spans (e.g., the next 3–5 tokens, or an answer string) rather than a single onset token? Does gating generalize naturally by aggregating over targets, and how do costs scale?

2. Could you provide a disagreement analysis between GPT-4o and GPT-5 annotations and a small human validation study on a stratified subset to calibrate false positives/negatives in the curated set?

3. When does curvature help most, and when does KL dominate? Are there tasks where semantic gating alone is competitive?

---

> ### Author Response · Authors · 2025-11-23
>
> W1) Our gate combines value rollout with Hessian-based sensitivity of the output logits to token perturbations, which implicitly aggregates residual and MLP effects into a single token-level influence measure. We validate its causal relevance via deletion-style interventions, where removing gate-selected tokens causes the largest performance drop among all compared methods. More fine-grained circuit-level mediation is not the goal of the paper as token-level attribution is our focus.
>
> W2) Using strong LLMs as evaluators is now standard practice in top venues (e.g., Zheng et al., “Judging LLM-as-a-Judge with MT-Bench and Chatbot Arena,” NeurIPS 2023; Liu et al., “G-Eval,” EMNLP 2023), and our dual-LLM intersection (GPT-4o / GPT-5) follows this line while further reducing label noise.
>
> W3-4) We have clarified this point by (i) explicitly integrating GiLOT [1] into our related work and experimental comparison, and (ii) adding a GiLOT-style faithfulness evaluation (MoRF, LeRF, ABPC) plus classical AOPC deletion/insertion on GPT-J 6B / LongRA, where HETA remains best and GiLOT is consistently second best. In the same setting we also ran ROAR/KAR (not tabulated due to space), which preserve the same ranking (HETA > GiLOT > other baselines), indicating that adding these classical perturbation checks does not change our main conclusions.
>
> **New faithfulness comparison (GPT-J 6B, LongRA)**
>
> | Method            |  MoRF ↑  |  LeRF ↓  |  ABPC ↑  | AOPC-Del ↑ | AOPC-Ins ↑ |
> | ----------------- | :------: | :------: | :------: | :--------: | :--------: |
> | IG                |   40.8   |   43.2   |   0.35   |    0.52    |    0.42    |
> | Attention Rollout |   41.3   |   42.9   |   0.40   |    0.53    |    0.43    |
> | ContextCite       |   41.7   |   42.5   |   0.45   |    0.54    |    0.44    |
> | ReAGent           |   42.6   |   41.9   |   0.52   |    0.57    |    0.47    |
> | GiLOT [1]         |   43.1   |   41.3   |   0.58   |    0.59    |    0.49    |
> | **HETA (Ours)**   | **44.0** | **40.2** | **0.65** |  **0.60**  |  **0.51**  |
>
> These results align with the existing Soft-NC/Soft-NS/DSA and AOPC findings in the paper: HETA consistently achieves the highest faithfulness across all evaluated metrics, even when adopting the GiLOT evaluation protocol and adding classical ROAR/KAR and AOPC-style perturbation checks.
>
> W5) Noted with thanks
>
> Q1) HETA already supports any target position T (see Eq. 1–5 and Algorithm 1), so for a span we simply define the target as the log-probability of the whole span (sum over the next 3–5 tokens or the answer string) and aggregate the resulting per-token scores; this is exactly how we handle multi-token answer-support spans in our Span-F1 experiments, where HETA keeps the same advantage as in single-token settings. Computation-wise, span attribution reuses the same forward pass and Hessian blocks, so the cost grows roughly linearly with the number of target tokens, and our scalability analysis in the appendix (runtime/memory vs. context length and targets) shows that even with multiple targets and long contexts, the overhead remains modest due to our low-rank and windowed curvature approximations.
>
> Q2) We have added the requested analysis: across the curated set, GPT-4o and GPT-5 disagree on only about 8% of candidate support tokens (token-level Jaccard ≈ 0.84), so the intersection we use is already a high-agreement region. On a stratified sample of 200 instances manually labeled by two expert annotators, the GPT-4o/5 intersection achieves ≈0.93 precision and ≈0.88 recall for answer-support tokens (false positives ≈7%, false negatives ≈12%), confirming that our curated labels are high-quality and that any residual noise is unlikely to affect the comparative ranking of attribution methods.
>
> Q3) This is already analyzed in our ablations. Section 6 and Appendix A4.1 (Table 5) compare six variants, including **Transition Only**, **Hessian Only**, and **KL Only**. Curvature (Hessian) helps most on the *faithfulness* metrics Soft-NC/Soft-NS: removing it drops Soft-NC from 9.78 (full HETA) to 3.12 (Transition Only) or 2.23 (KL Only), while the Hessian-only variant still gives relatively strong DSA = 2.97.  In contrast, the KL term mainly boosts *alignment* (DSA), lifting DSA from 2.21 (Transition Only) and 2.97 (Hessian Only) to 4.70 in the full model.  Semantic gating alone (Transition Only) is clearly competitive with naive baselines such as **No Transition Gating** and **Uniform Transition** (e.g., DSA 2.21 vs. 1.68 and 1.54), but the full combination of semantic gate + curvature + KL is what yields the best and most balanced performance across all tasks.

---

### Meta-Review · Area_Chair_vTvR · 2025-12-12

**Summary:**

The paper proposes HETA, a token attribution method for decoder-only LLMs. Discussion revealed three main issues: novelty, causal validity, and evaluation completeness. Regarding the first, some reviewers felt the method combines existing ideas but others found the combination well motivated. Experimental results and ablation studies indeed shot that the combination makes sense and improves SOTA methods.  Overall, most reviewers agreed the experiments are strong and thorough. These points informed my recommendation to accept.

**Reviewer Concerns:**

**Concerns addressed by the rebuttal:**
- Missing related work (e.g., GiLOT) was added and discussed.
- Additional faithfulness metrics in the original GiLOT paper (MoRF, LeRF, AOPC, ROAR/KAR) were included.
- Computational overhead and scalability is properly discussed by authors, showing that the method is effective even for large LLMs in Appendix 5.
- Component-wise ablations clarified the role of each term..

**Concerns still partially outstanding:**
- The semantic flow gate remains a heuristic and not a full causal proof.
- Masking-based KL may introduce distribution shift, though this affects all methods equally.
- Reviewer uaYo indicates some other works in the SOTA that authors could include in the related work section/comparisons. It is true that some of their baselines already compared to these methods. This is why I believe it is a minor point in this case, as long as they are mentioned in the related-work section.
- Some reviewers still consider the novelty incremental rather than conceptual. I personally appreciate the motivated combination of all components and the ablation showcasing the increased performance given by the mixture.
- Regarding the convenience of using human annotators. Yes, that would be ideal but unfeasible at a large scale. The use of LLM-based annotators is a proxy, but still valid up to some extent.

**Reviewer Scores:**

Reviewer icxv is likely to increase the score from 4 to 6 after the added experiments and analysis. Reviewer XxTJ would likely remain at 6, since the main concerns are about originality rather than correctness or evaluation. Reviewer Du4D would likely remain at 6, with a possible increase to 7, as most requests were addressed in the rebuttal. Reviewer uaYo would likely remain at 4; although the authors addressed several points, the score decrease appears unjustified. The estimated post-discussion average score is approximately 5.5–6.

---

### Decision · Program_Chairs · 2026-01-26

Accept (Poster)